# Timely bespoke phage-antibiotic combination to treat refractory *Pseudomonas aeruginosa* mediastinitis and vascular graft infection

Biofilm-related vascular graft infections (VGIs) pose major therapeutic challenges due to persistent, antibiotic-resistant bacteria often residing in retained grafts. Phage therapy offers a promising alternative treatment strategy against biofilm-associated infections, though its use remains mostly ad hoc and typically considered a last-resort intervention. We report here the treatment of a refractory, fluoroquinolone non-susceptible *Pseudomonas aeruginosa* VGI using a systematically planned and synergistic phage-antibiotic combination approach. Adjunctive phage therapy led to radiological improvement, as seen by reduced 18F-FDG PET/CT tracer uptake around the graft. The patient was transitioned to oral fluoroquinolone suppression therapy with no recurrence of bacteremia to-date, after a year. Our workflow led to the selection of phages that sensitized *Pseudomonas aeruginosa* to killing by levofloxacin and piperacillin-tazobactam. We established that this phage-driven antibiotic sensitization was due to the ability of our phages to use the MexAB-OprM efflux pump as a receptor. We also showed that our phages had potent anti-biofilm activity. We advocate a systematic, multi-pronged management strategy for refractory VGIs, including early therapeutic drug monitoring (TDM), in vitro antibiotic combination testing (iACT), and timely adjunctive phage therapy. This case illustrates the utility of individualized, strategic approaches and highlights adjunctive phage therapy's potential in treating complex biofilm-related infections.

Vascular graft infections (VGIs), although uncommon, are serious infections following arterial reconstruction surgeries. VGIs are associated with significant morbidity and mortality, especially in patients with intra-cavity VGIs[1–3]. Definitive treatment with graft explant is not always possible due to patient factors and surgical technical challenges, especially in patients with composite valve grafts (e.g., ascending aortic graft infections)[3]. If the graft is preserved, treatment of the VGI with a prolonged course of parenteral antibiotics followed by long-term suppressive antimicrobial therapy is often recommended

due to the presence of biofilm; survival rates are low and infection recurrences common[4,5]. In refractory VGIs, the infected grafts are often left in situ; the causative bacteria often form drug-tolerant biofilm, and the emergence of persistent bacteria subpopulations is not unexpected[6,7]. With time and ongoing antibiotic selection pressure, VGIs are likely to become drug resistant, and harder to treat.

The development and early application of biofilm-targeting novel therapeutic strategy is crucial for the treatment of refractory VGIs; adjunctive bacteriophage (phage) therapy serves this purpose, and

✉e-mail: wilfried.moreira@ntu.edu.sg; andrea.kwa.l.h@sgh.com.sg

interest in this field is growing[8–10]. Phages are viruses that specifically infect bacterial cells and can cause bacterial lysis. Using phages for treatment offers several advantages[9]. Phages have been shown to be active against both planktonic and biofilm-forming bacteria. Phages have also been shown to enhance antibiotic activity by eliciting phage-antibiotic synergies and restoring susceptibilities of otherwise antibiotic-resistant bacteria to antibiotics[10]. In a targeted review exploring the role of phages (used in combination with antibiotics) in VGIs, bacterial eradication was achieved in 6 out of 7 (86.6%) cases where either *Staphylococcus aureus* and/or *Pseudomonas aeruginosa* (*P. aeruginosa*) were isolated, both notorious for biofilm formation. Clinical improvement was seen in 4 out of 7 (57.1%) cases[9].

A 36-year-old female patient with a history of congenital heart disease (pulmonary atresia with ventricular septal defect) with 4 prior sternotomies for corrective heart surgery developed progressive dilatation of her ascending aorta with severe aortic regurgitation and pulmonary homograft stenosis in 2023. This was associated with admissions for decompensated heart failure. To address this, she eventually underwent a Bentall procedure where the affected aortic valve, aortic root and ascending aorta were replaced by a composite vascular graft and the pulmonary homograft was also replaced in the same setting on January 25th, 2024, just 5.5 weeks after she had delivered her second child. Surgery was technically challenging due to dense adhesions from prior sternotomies with significant intra-operative bleeding, and inadvertent tear in the aorta during adhesiolysis for which gelweave patch augmentation of the inferior aspect of the aortic arch was performed.

Her post-operative course was eventful and protracted (See Fig. 1). She developed a febrile illness on post-operative day (POD) 2 and was treated for culture-negative healthcare associated infection with a 5-day course of piperacillin-tazobactam with response. However, on POD 9, she developed drug sensitive *P. aeruginosa* mediastinitis. (See Fig. 2A.) Mediastinal washout performed on POD 15 showed good clinical and biochemical response. When her condition stabilized, she was transitioned to oral ciprofloxacin 750 mg, completing 8 weeks of antibiotics, with normalization of white blood cell (WBC) $8.32 \times 10^9$/L and C-reactive protein (CRP) was 11.2 mg/L at end of therapy.

However, within 4 days of discontinuing antibiotics, her fever recurred. She only presented to care 10 days later with ongoing fever and chills with an elevated, WBC 16.71 $\times 10^9$/L, CRP 283 mg/L and procalcitonin 12.4 µg/L. She also suffered a high-grade *P. aeruginosa* bacteremia which had evolved to be ciprofloxacin non-susceptible. Although the CT scan showed no signs of recurrent mediastinal abscess, metastatic infection, or infective endocarditis, there were significant concerns about the progression of refractory or resistant intra-cavity thoracic VGIs. Due to the high surgical risk, a redo Bentall procedure was considered unsafe.

In view of the above, a multi-disciplinary team was convened, and we opted for a systematic and individualized approach to manage the *P. aeruginosa* VGI. Tailored antibiotic therapy, guided by therapeutic drug monitoring (TDM) and in vitro antibiotic combination testing (iACT), was implemented. Additionally, personalized phage therapy was considered early and integrated into the treatment plan. This resulted in the timely administration of salvage phage therapy and restoration of oral fluoroquinolone efficacy supporting chronic suppression of the VGI.

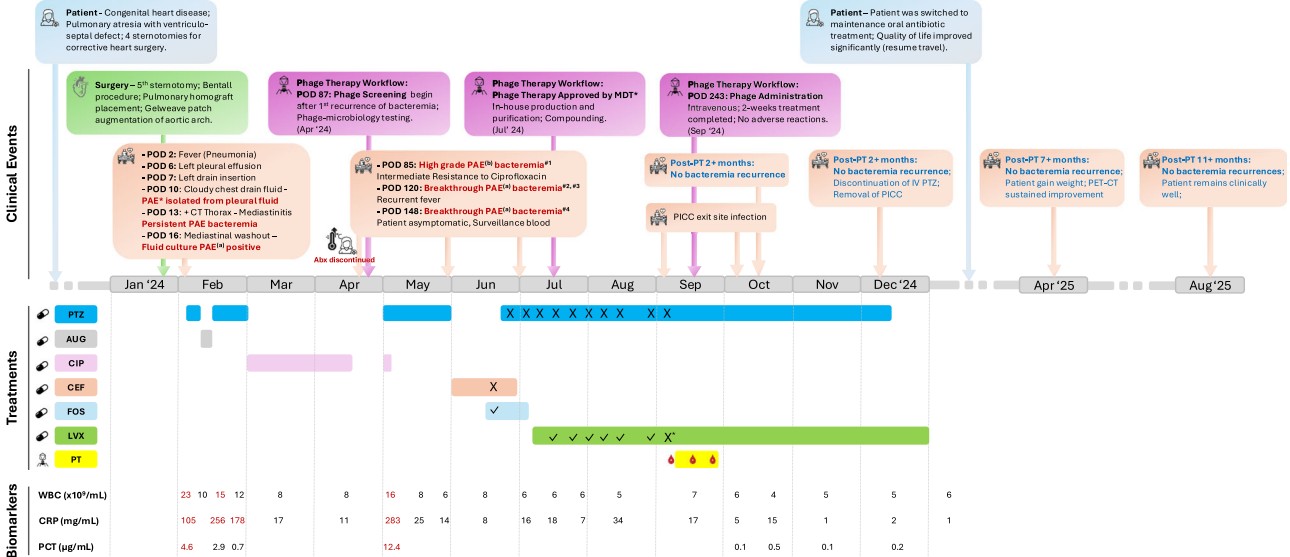

**Fig. 1 | Clinical course of patient with refractory *Pseudomonas aeruginosa* vascular graft infection.** The patient experienced recurrent *Pseudomonas aeruginosa* bacteremia despite appropriate culture-directed antibiotic therapy. Therapeutic drug monitoring (TDM) indicated that standard dosing achieved subtherapeutic drug levels for the deep-seated vascular graft infection (VGI) (details in Supplementary Information), likely contributing to recurrent−though progressively less symptomatic−infections. In late June 2024, combination therapy with high-dose piperacillin−tazobactam (27 g/day) and oral levofloxacin (750 mg daily) cleared the bacteremia. Adjunctive intravenous phage therapy was administered for two weeks in late September 2024 to target the suspected biofilm-associated component of the infection. During prolonged intravenous antibiotic therapy, recurrent PICC line−related exit site infections occurred. Given sustained clinical and radiological improvement, intravenous antibiotics were discontinued in December 2024, and the patient transitioned to oral levofloxacin monotherapy for VGI suppression. The patient remains clinically well on follow-up. Clinical events abbreviations: POD post-operative days, PAE *Pseudomonas aeruginosa*, CT computed tomography, MDT multi-disciplinary team, PICC Peripherally Inserted Central Catheter, Post-PT Post-phage therapy, IV PTZ intravenous Piperacillin/Tazobactam, PET-CT Positron Emission Tomography (PET) and Computed Tomography (CT); a Fluoroquinolone-susceptible *Pseudomonas aeruginosa* isolates; b Fluoroquinolones-non-susceptible (intermediate resistance) *Pseudomonas aeruginosa* isolates. Treatments abbreviations: PTZ Piperacillin/Tazobactam, AUG Augmentin, CIP Ciprofloxacin, CEF Cefepime, FOS Fosfomycin, LVX Levofloxacin, PT Phage Therapy, Red blood drop symbols during PT, indicate dates at which blood sample were taken to determine phage presence in patient blood (day 0, 8 and 15). X, therapeutic drug monitoring (TDM) showed inadequate sub-therapeutic concentration of antibiotics; √, Therapeutic drug monitoring showed adequate therapeutic concentration of antibiotics; X*, The dose of levofloxacin was reduced to 500 mg due to nausea, but levels were deemed inadequate, so after the TDM was made available, levofloxacin dose increased back to 750 mg daily. Biomarkers abbreviations: WBC white blood cells, CRP C-reactive protein, PCT Procalcitonin.

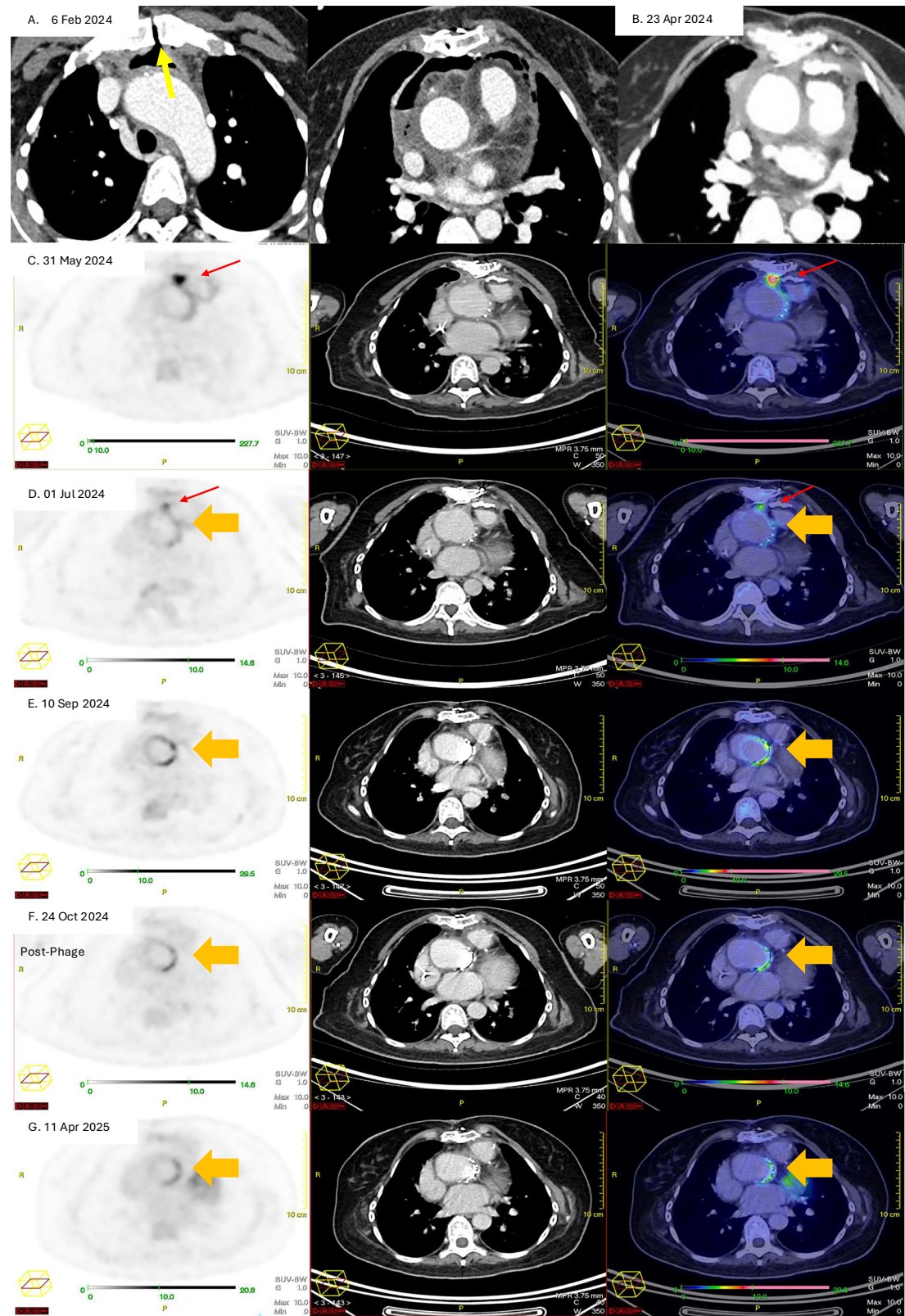

## Results

### Systematic approach – Optimizing antibiotic regimen and initiating phage workflow

In mid-April 2024, the patient was reinitiated on treatment for recurrent *P. aeruginosa* bacteremia. Biomarker response to the piperacillin-tazobactam 4.5 g q6hly, breakthrough fever and recurrent bacteremia occurred quickly within 5 weeks of therapy. In response, TDM was performed to guide antibiotic dosing and improve care (Table S1). Antibiotics dosages were adjusted based on attainable drug levels for a deep-seated infection, biofilm activity, "perceived" or proven synergism in vitro, as well as feasibility of outpatient antibiotic administration. (Fig. 1. Details are described in the Supplementary Information).

**Fig. 2 | Serial computed tomography and positron emission tomography computed tomography images illustrating the progress of the vascular graft infection before and after phage therapy. A** CT Aortogram (Thorax) on 6 February 2024, following her Bentall procedure, showing heterogeneous anterior mediastinal collection with gas locules surrounding the aortic arch and ascending aorta. There is gas seen tracking through the midline sternotomy (yellow arrow). **B** CT Aortogram on 23 April 2024 shows resolution of the pneumomediastinum but there is interval development of a hyperdense rind along the ascending aorta and within the anterior mediastinum. **C–E** Pre-phage therapy attenuation corrected 18F-FDG PET image. CT image and fusion 18F- FDG PET/CT images at different time points in chronological order. There is an initial intense 18-FDG-avid focus of uptake posterior to the sternotomy wire (red arrow). This shows interval decrease in intensity and then subsequent metabolic resolution by 10 September 2024. However, there is serial interval increase in the circumferential 18F-FDG uptake around the aortic valve prosthesis (orange arrow), suggestive of an underlying active infective/inflammatory process. **F,G.** Post-phage therapy attenuation corrected 18F-FDG PET image. CT image and fusion 18F-FDG PET/CT images at different time points in chronological order. The most intense tracer uptake around the graft was seen in September 2024, and following phage therapy, there is interval sustained improvement following the administration of phage therapy showing interval decrease intensity of the circumferential 18F-FDG uptake along the aortic valve prosthesis (orange arrow).

With standard and even high dose drug administration with prolonged infusion, the serial β-lactam drug levels were deemed inadequate for the treatment of deep tissue infection(s) (detailed Supplementary Information). It was therefore not surprising that recurrent breakthrough of bacteremia occurred (three episodes in 3 months). The patient was subsequently placed on high-dose piperacillin-tazobactam 27 g/day and oral levofloxacin 750 mg daily in June 2024 based on synergy E-test results and the bacteremia cleared. (See the Supplementary Information.)

Given the refractoriness of the VGI, a multi-disciplinary team comprising infectious diseases physicians, microbiologists, pharmacists, treating cardiothoracic surgeons, and phage researchers was convened in June 2024, and consensus recommendations for compassionate use of adjunctive phage therapy was made. Thereafter, a collaborative effort within the Singapore Phage Network was initiated.

The decision to pursue adjunctive salvage phage therapy was driven by her clinical course and serial 18F-fluorodeoxyglucose positron emission tomography/computed tomography (18F-FDG PET/CT) findings, which showed persistent tracer uptake in the aortic graft despite prolonged high-dose combination antibiotic therapy guided by TDM. However, the risk of relapse once antibiotics were discontinued was concerning. (See Fig. 2.) In addition, there were interval development of vascular access complications: Vascular catheter exit site infection developed on September 4, 2024, necessitating line replacement; long term intravenous antibiotic therapy was not tenable. Definitive surgery to remove the composite vascular graft was technically challenging with high operative risks, significant morbidity and mortality, which was unacceptable to the patient. Phage therapy offered the possibility of restoring antibiotic susceptibility to fluoroquinolones and beta-lactams with the aim to suppress the VGI. Approval for the use of investigational phage therapy was obtained from our hospital medical ethics committee. The patient was also counselled, and she later consented to receiving phage-antibiotic combination therapy as part of salvage treatment.

## Microbiological profiles

With the first recurrence of *P. aeruginosa* bacteremia in April 2024, the bacterial isolates from this patient were sequentially archived and used to screen our phage repository, as well as to evaluate phage-antibiotic combination, and to produce selected phages for compounding of the selected therapeutic cocktail. A total of 4 blood culture *P. aeruginosa* isolates (PA2081, PA2091, PA2096, and PA2111) were retrieved between April and June 2024. Despite morphological differences, they exhibited similar antibiotic susceptibility profiles (see Tables S2, S3). All four isolates belonged to ST252 and shared the similar resistance/ virulence gene profiles (Table S4 in the Supplementary Information). Pairwise genomic alignment has reflected a highly conserved profile with genomic similarity between 98.3% to 99.8% across isolates. The prophages (or Phage Defense System) in these 4 isolates were identified and characterized (Table S4 and Fig. S1). Prophages are devoid of VF, AMR genes or other remarkable genetic elements.

## Identification of suitable therapeutic phages

The Singapore Phage Repository—established through collaboration between Singapore General Hospital, Singapore Eye Research Institute, National University of Singapore, and Nanyang Technological University—was systematically assembled to include therapeutic-grade phages with potent lytic activity against clinically relevant pathogens. Phages were isolated using well-characterized clinical isolates of *P. aeruginosa*, *Klebsiella pneumoniae*, *Acinetobacter baumannii*, and *Staphylococcus aureus*, representing prevalent sequence types and resistance profiles from Singapore General Hospital and other local institutions. All phages underwent whole-genome sequencing (WGS) and annotation to confirm their lytic nature and exclude those carrying antimicrobial resistance genes, virulence factors, or lysogeny-associated markers. Only strictly lytic, genomically "clean" phages were retained, ensuring a diverse, well-characterized collection suitable for precision phage therapy. This repository served as the primary phage source for this case.

## Identification of Phage Candidates and Selection of Therapeutic Phage Cocktail

A multidimensional evaluation workflow (Fig. 3) was used to identify the most suitable therapeutic phages from the Singapore Phage Repository. Screening began with phage susceptibility testing of the four *P. aeruginosa* clinical isolates using spot and plaque assays. Eight distinct phages demonstrated infectivity (Figs. S2, S3). Lytic activity was quantified through time-kill assays, assessing initial bacterial killing, regrowth kinetics, and final bacterial counts at 24 h. Five single phages, PW20, PW21, PW23, KSY1a, and P0413, were shortlisted based on diverse killing profiles and selection pressures (Figs. S4–S7).

From these candidates, we evaluated 20 phage–phage combinations (two- and three-phage cocktails), identifying potential phage-phage synergistic activities. Four 3-phage cocktails exhibited superior performance, achieving greater initial killing, sustained suppression of bacterial regrowth, and delayed rebound compared with single phages or 2-phage cocktails. However, none achieved complete eradication without antibiotic co-administration (Figs. S8, S9).

Phage Cocktail #3, comprising PW21, KSY1a, and P0413, was selected for therapeutic use as it demonstrated consistent high potency against both planktonic and biofilm cultures of all four *P. aeruginosa* clinical isolates, robust synergistic activity with key antibiotics (e.g., tazobactam, levofloxacin) (Fig. 4A–D, Table S9, and Figs. S10–S16), and broad coverage with minimal cross-resistance (Table S10). In addition, all three constituent phages were genomically confirmed as strictly lytic and free of undesirable genes and remained available in high-titer stocks suitable for intravenous administration.

## Phage-Driven Antibiotic Sensitization

Phage-driven antibiotic sensitization was identified by assessing minimum inhibitory concentration (MIC) shifts and enhanced bacterial killing in combination assays. MIC testing of the four patient isolates revealed substantial reductions when antibiotics were combined with Phage

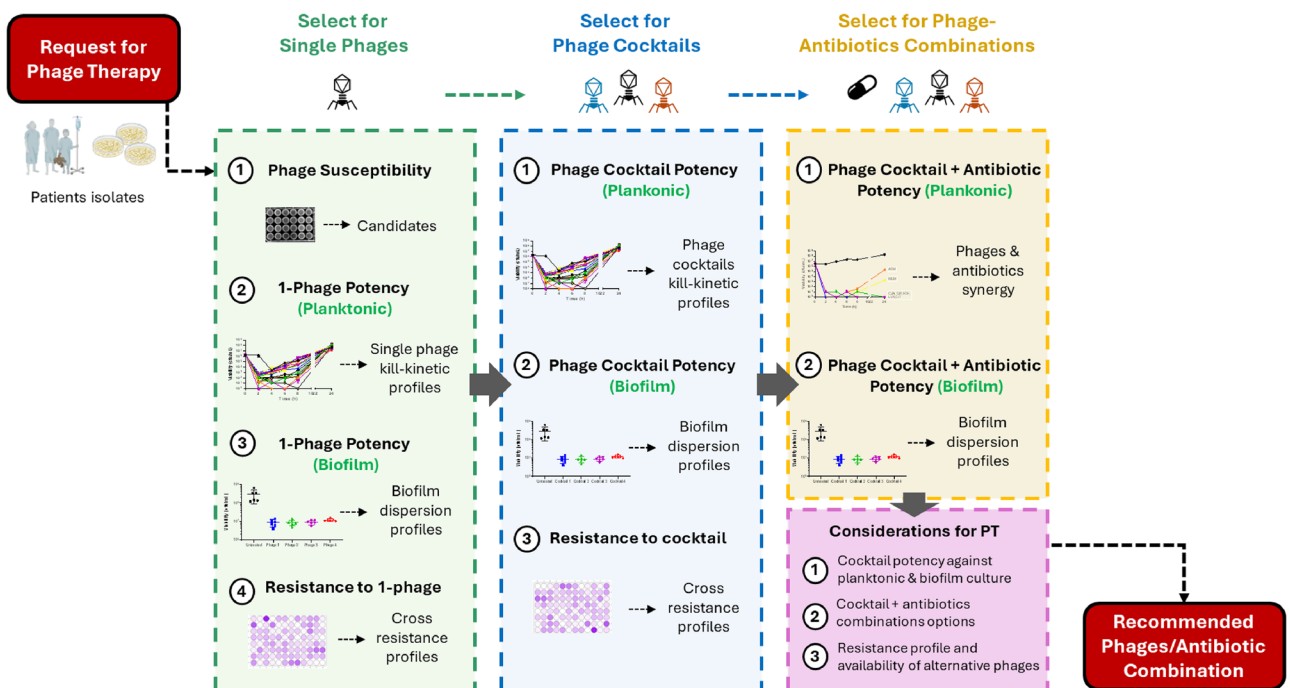

**Fig. 3 | Singapore Phage Therapy Workflow.** Workflow diagram showing the phage therapy development process, from "Request for Phage Therapy" through phage selection steps (for phage cocktails and phage-antibiotic combinations) to the final "Recommended Phage/Antibiotic Combination." Our workflow starts with the collection of patient clinical isolates and follows a three-steps process: selection of single phages, selection of phage cocktails and selection of phages/antibiotics combinations. Single-phage selection starts with a phage-susceptibility testing of the patient isolates against our phage library. The potency of phage candidates is next determined against planktonic and biofilm cultures of the patient isolates using viability (CFU enumeration) and biomass assays (crystal violet). Single-phage escape mutant (bacteria resistant to phage infection) are collected and cross-resistance to other phages is determined. Phages cocktails (composed of 2 or 3-phages) are evaluated in a similar manner. Finally, treatments of cocktails in combination with different antibiotics are screened. The choice of the combination of a phage cocktail with antibiotic take into consideration three primary criteria: potency, antibiotics options and our ability to counter the emergence of resistance with alternative combinations.

Cocktail #3, often restoring susceptibility (Fig. 4A–C). Kill-kinetic and growth inhibition assays demonstrated that the combination of cocktail #3 (individual phages or as a cocktail) and piperacillin–tazobactam or levofloxacin, achieved sustained killing below the limit of detection at MIC concentrations and enhanced inhibition at sub-MIC levels (Fig. 4A–D). Viability and biofilm biomass assays further confirmed that phage cocktails not only dispersed mature biofilms but also ensured bacterial killing (Fig. 4E). We also established that PW20, PW21 and PW23 rely on MexAB-OprM membrane proteins to infect *P. aeruginosa*. The ability of PWs phages to utilize the MexAB-OprM efflux pump as a receptor provide the foundation for the underlying mechanism of phage-driven antibiotic sensitization (Fig. S17). The genomic profiles of the three therapeutic phages selected (PW21, KSY1A, and P0413) are provided in the Supplementary Information (Tables S5–S8). PW21 and P0413 (both Myoviridae) have genome lengths of 67,991 bp and 66,492 bp, respectively, while KSY1a (Phikzvirus) has a significantly larger genome of 282,489 bp, which makes it a jumbo phage.

### Intravenous phage treatment
Phage therapy administration and monitoring was adapted from the STAMP protocol[11]. A 2-week course of purified 3-phage cocktail, PW21, P0413 and KSY1a were selected, and phage titers between $10^8$ and $10^9$ plaque-forming unit (PFU) was finally administered (daily on the first 2 days, and twice daily for the remaining 12 days) in combination with her existing antibiotic regimen (piperacillin-tazobactam and levofloxacin) on September 23, 2024.

### Clinical course following phage therapy
The patient tolerated phage therapy well with no adverse events observed during treatment. Biochemically, we noted a

measurable increase in the inflammatory markers; the previously normal CRP and erythrocyte sedimentation rate (ESR) increased to a peak of 16.8 mg/L, and 44 mm/h, respectively, after phage administration.

The patient was symptom free before and after phage therapy with no impairment in daily activities. The most significant observation following phage administration was the reduction in tracer uptake around the prosthetic aortic valve 1 and 2 months after phage therapy on serial 18F-FDG PET/CT (Fig. 2F). The serial CRP and ESR readings also returned to normal limits 15 days, and 85 days post phage therapy, respectively. Intravenous piperacillin-tazobactam was discontinued on December 3, 2024, and the central venous catheter was removed. The patient was subsequently maintained on levofloxacin monotherapy with no recurrence of bacteremia.

### Incorporating systematic therapeutic monitoring during phage therapy
Here, we performed therapeutic phage monitoring, which is clinical monitoring and evaluation of patients while undergoing phage therapy, to 1) assess safety profile, specifically to identify unwarranted, adverse immune reactions triggered by phage administration; 2) detect phages in patients. Phages are nucleoprotein complexes which are potentially highly immunogenic, particularly when administered intravenously. Clinically, the patient exhibited no signs or symptoms indicative of systemic inflammation or anaphylactic responses. However, slight increases in CRP and ESR, as previously noted, alluded to a mild inflammatory response post phage administration. Phage were detected in patient blood samples at day 8 and day 15 of phage therapy by PCR using primers designed to discriminate between the three phages (Figs. S18, S19 and Table S11).

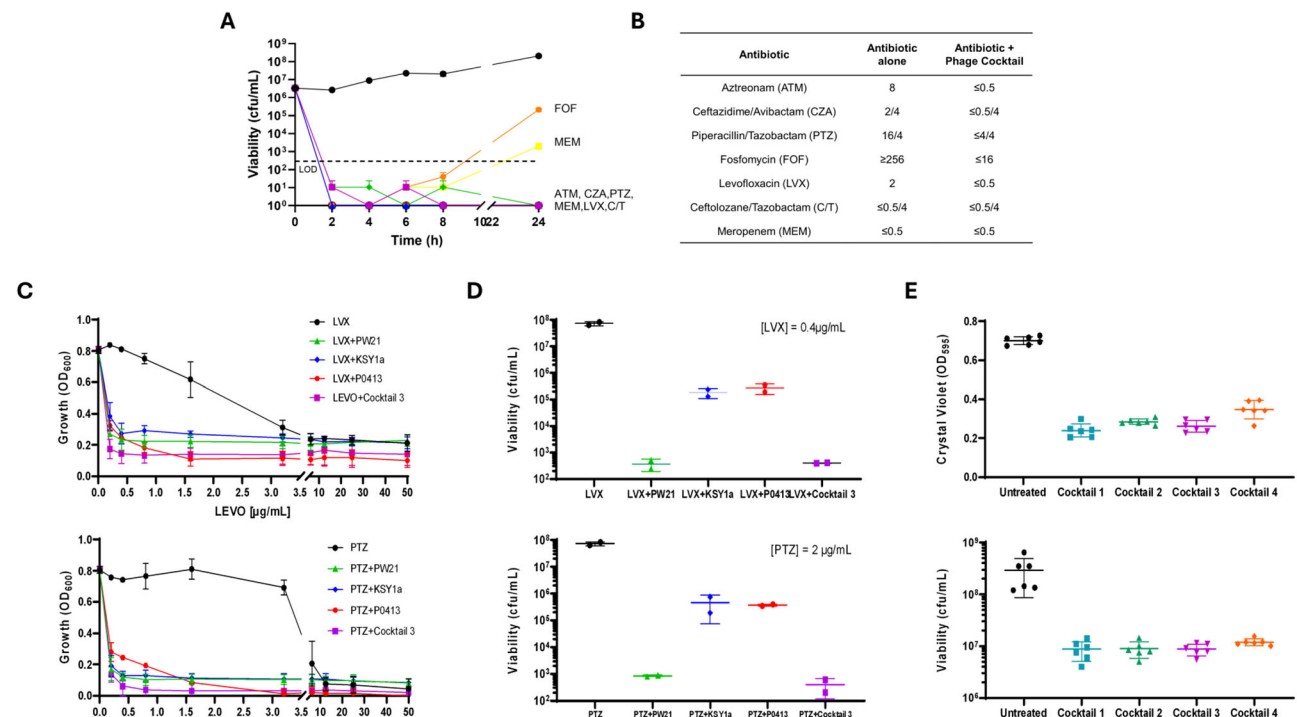

**Fig. 4 | Therapeutic Phage Cocktail Drives Antibiotic Sensitization. A** Kill-kinetic of phage cocktail #3 (phages PW21, KSY1a and P0413) in combination with antibiotics. The combination of phage cocktail #3 combined with CZA, PTZ, MEM, LVX and C/T leads to sustained killing (below limit of detection, LOD) over time. Antibiotics were used at MIC concentration, phage cocktail #3 was used at MOI 100. **B** MIC table of the 4 patient isolates showing reduction in MIC when antibiotics are used in combination with phage cocktail #3. **C** Growth inhibition of the patient clinical isolates upon treatment with single-phages or phage cocktail #3 combined with LVX (upper panel) or PTZ (lower panel). The combination of phages (alone or as a cocktail) induce a reduction in MIC. **D** Viability assay of the patient isolates treated with sub-MIC concentration of LVX (upper panel) or PTZ (lower panel) in combination with single-phages or phage cocktail #3. The adjunction of phages (alone or as a cocktail) to LVX or PTZ leads to an increase in sub-MIC killing.

**E** Biofilm assay measuring biomass (crystal violet staining, upper panel) and viability (CFU enumeration) after treatment with phage cocktails (cocktail #1: PW20, PW21, KSY1a, cocktail #2: PW21, PW23, KSY1a, cocktail #3: PW21, KSY1a, P0413 and cocktail #4: PW23, KSY1a, P0413). All three cocktails induce dispersion and killing of mature (24 h) biofilm. The decision to combine all 4 bacterial isolates and use a high MOI of 100 for phages was driven by the clinical context of a polymicrobial, biofilm-associated infection involving genetically related but phenotypically variable strains. This approach aimed to simulate in vivo complexity and ensure maximal killing across the population. Data are presented as mean values +/- SD of 2 biological replicates. Source data are provided as a Source Data file. Abbreviations: ATM Aztreonam, CZA ceftazidime/avibactam, PTZ piperacillin/tazobactam, FOF fosfomycin, LVX levofloxacin, C/T ceftolozane/tazobactam, and MEM meropenem.

## Discussion

Management of refractory, biofilm-associated infections requires a multi-modal, multidisciplinary strategy. Despite months of high-dose, TDM-guided antibiotic therapy with iACT optimization, serial 18F-FDG PET/CT scans demonstrated persistent tracer uptake on the aortic graft, suggesting likely long-term treatment failure. Explantation of the infected graft was deemed infeasible due to prohibitive surgical risk from multiple prior sternotomies and procedural complexity. Standard β-lactam dosing yielded suboptimal serum concentrations and recurrent bacteremia, underscoring the importance of early TDM-guided, personalized dosing even in non-critically ill patients with difficult-to-treat (DTT) infections[12–15]. When therapeutic levels were not achievable, iACT enabled identification of additive antibiotic combinations[16,17], in this case, piperacillin–tazobactam plus levofloxacin[18]. While bacteremia was briefly controlled (in accordance with iACT and clinical outcomes[19]), the chances of treatment failure remained high.

Adjunctive phage therapy was considered early (i.e., before deterioration) in April 2024 because <10% of referred cases receive treatment, active products are available for only 30–50% of eligible patients, and procurement typically takes months[20,21]. Outcomes from tailored phage therapy are promising, with ~80% clinical improvement and low adverse event rates (< 10%)[22–24]. In our setting, leveraging TDM and iACT[25,26], alongside a pre-existing phage infrastructure (well

characterized phage repository, susceptibility testing, synergy assays, biofilm models, and Good Manufacturing Practice [GMP]-like production) proved critical, as recognized by global phage therapy efforts[21,27–30], and enabled rapid screening and delivery of a therapeutic product, marking Singapore's first reported clinical phage use. This report proposes a framework for the timely systematic incorporation of compassionate adjunctive phage therapy for DTT infections (Fig. 5.)

The therapeutic cocktail (PW21, KSY1a, P0413) was chosen after systematic screening of our repository against all four *P. aeruginosa* clinical isolates. Each phage cocktail demonstrated potent lytic activity in planktonic cultures and robust biofilm-disrupting capacity, significantly reducing both biomass and viable cell counts in mature biofilms. Importantly, the combination retained efficacy across isolates with differing resistance phenotypes.

The cocktail showed marked synergy with key antibiotics, as evidenced by substantial MIC reductions for piperacillin–tazobactam and levofloxacin, and increased killing at sub-MIC concentration. Time–kill studies demonstrated sustained bactericidal activity of the phage–antibiotic combinations, far exceeding the inhibitory effects seen in MIC assays alone. This enabled the use of oral levofloxacin for long-term suppression, critical for outpatient management and improved quality of life. As the patient has not had any infection relapse to-date post phage therapy, and resistance profiling of isolate post treatment with phage cocktail in kill kinetic study is not available,

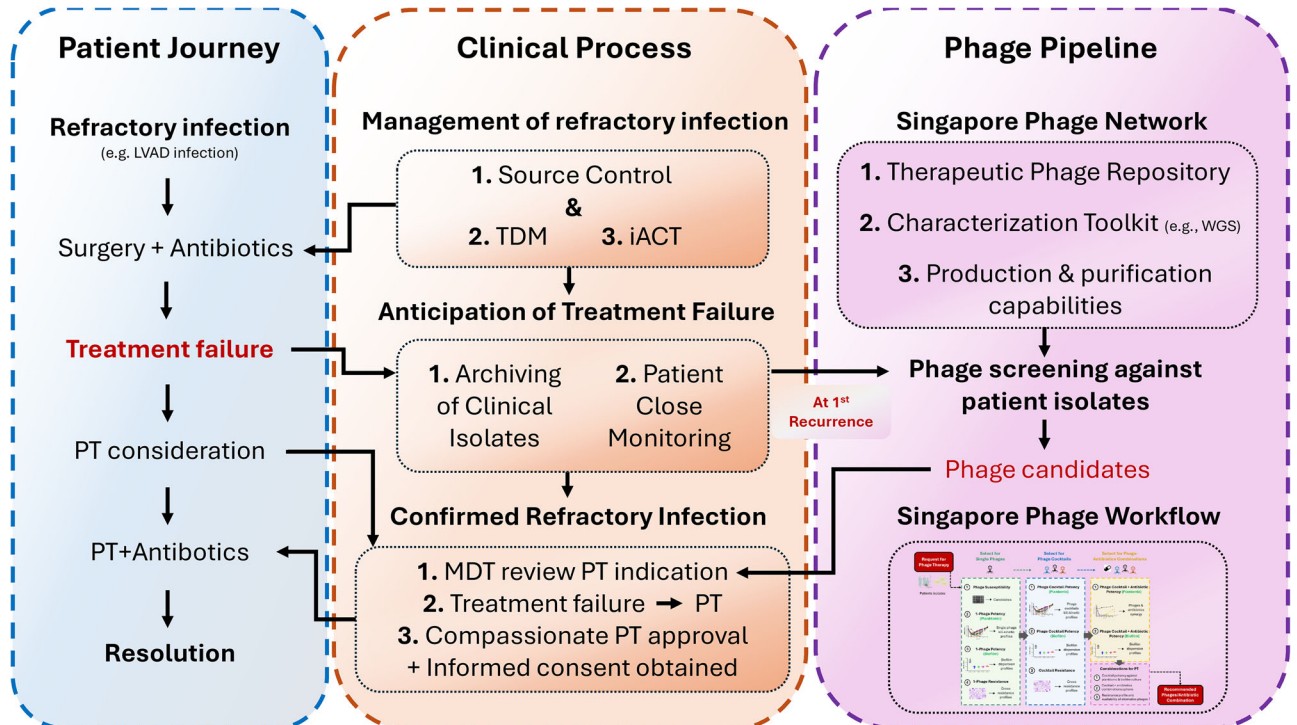

**Fig. 5 | Multi-modal, multi-disciplinary integrated workflow for managing refractory infections and implementing phage therapy.** The diagram illustrates three interconnected components: **Patient Journey (left panel)** – Progression from initial illness (e.g., LVAD infection) through surgery and antibiotics, possible treatment failure, consideration of phage therapy (PT) combined with antibiotics, and resolution. **Clinical Process (center panel)** – Management of refractory infection through source control, therapeutic drug monitoring (TDM), and individualized antibiotic combination testing (iACT); anticipation of treatment failure via archiving of clinical isolates and close monitoring; and confirmed refractory infection triggering multidisciplinary team (MDT) review, compassionate-use approval, and informed consent. **Phage Pipeline (right panel)** – The Singapore Phage Network supports therapeutic phage discovery and application through a repository, genomic and phenotypic characterization tools, and production capabilities. Patient bacterial isolates undergo phage screening to identify candidates, which then enter the Singapore Phage Workflow (described in Fig. 3) for testing, purification, and preparation for therapeutic use. Arrows depict how early phage screening can occur in parallel with antibiotic optimization to enable timely phage–antibiotic combination therapy.

we acknowledge the need of post-treatment isolate characterization and genomic follow-up as a critical component for future studies to confirm the true reversion of resistance.

Mechanistically, PW21 was found to exploit the MexAB–OprM efflux pump as its receptor, a feature that directly links phage infection to increased antibiotic susceptibility by altering efflux capacity[31–34]. This unique features of our adjunctive phage therapy approach positively impacted the antibiotic options for our patient, potentially allowing de-escalation of antibiotics, discontinuation of intravenous therapy, and improvement in patients' quality of life. In particular, fluoroquinolone susceptibility was restored, resuscitating its use as an oral suppressive antibiotic for the long-term management of the VGI.

Phage therapy was well tolerated, with only transient biochemical inflammatory responses and no clinical hyperinflammation. While these transient inflammatory responses were not explored here, further immunogenicity studies are needed, particularly regarding repeat dosing and neutralization. PCR confirmed phage persistence in blood at days 8 and 15, though in vivo amplification could not be demonstrated due to limited sampling. At 12 months post-therapy, the patient remained relapse-free on oral suppression, with improved PET/CT findings, weight gain, and restored quality of life. We are mindful of the preponderance of successful reports, and are aware of the potential for phage therapy to fail, as cautioned by Aslam et al.[35] when phages were used in similar cases to treat *P. aeruginosa* cardiac device-related infections.

This single-patient, compassionate-use case limits the ability to definitively attribute clinical improvement to phage therapy, as

concurrent optimized antibiotic therapy and surgical source control may also have contributed. The small number of post-treatment samples precluded robust in vivo phage amplification studies, and while PCR confirmed phage persistence, direct evidence of replication was lacking. Similarly, our serum neutralization assay data were inconclusive. The use of pooled bacterial isolates and a high multiplicity of infection (MOI) during in vitro synergy testing, while expedient in a time-sensitive context, may not fully reflect in vivo dynamics. Future work should incorporate standardized synergy testing at clinically relevant bacterial densities, longitudinal isolate collection for genomic follow-up, and systematic immune monitoring to assess neutralization and host responses. Development of streamlined workflows for rapid phage–antibiotic screening, including sub-MIC testing and biofilm assays, will further enhance the precision and translational value of therapeutic phage selection.

Before phage therapy, hospitalization costs reached $150,000 over 40 days, with projected lifelong outpatient intravenous antibiotics ($6,000/month) that accompanied by recurrent line-related complications post discharge. Without phage therapy, escalating cost from progression of the VGI (potentially with more resistant *P. aeruginosa*), further hospitalizations (with possible need for care in the intensive care unit should severe sepsis set in), and line-related complications are expected. Phage therapy enabled intravenous cessation after ~10 months (including 9 months of outpatient antibiotics, ~ $44,000), significantly reducing direct costs, morbidity and the burden of long-term antimicrobial use.

This case highlights the need for access to updated, well-characterized phage libraries matched to regional pathogens, rapid

screening against patients isolates and clinical phage microbiology capacity (susceptibility, synergy, and biofilm testing). Timely phage therapy also requires GMP or GMP-like production and post-treatment genomics to track resistance, virulence, and fitness changes.

Our experience supports the feasibility and clinical value (with potential health system benefits, especially in healthcare cost avoidance) of integrating phage therapy early into the management of DTT biofilm-associated infections, especially when supported by local infrastructure for rapid screening and production.

# Methods

## Ethical statement

Informed consent for the compassionate use of phage therapy was obtained from the patient and prior approval to proceed was sought from the SingHealth Translation Medicine Office. According to the CARE guidelines and in compliance with the Declaration of Helsinki principles, the therapeutic monitoring and clinical outcome assessment of adjunctive phage therapy was approved by the SingHealth Centralized Institutional Review Board (CIRB 2023/2415). Written informed consent for publication of clinical details and clinical images was obtained from the patient. A copy of the consent form is available for review by the Editor of this journal.

## Sample and isolate collection

Bacterial isolates were recovered from routine clinical cultures of the patient blood. *P. aeruginosa* isolates were sequentially collected before phage therapy. All isolates were identified by matrix-assisted laser desorption ionization-time of flight (MALDI-TOF) mass spectrometry (BrukerDaltonics, Billerica, MA, United States) using in-vitro diagnostic (IVD) library revision K (2020) software following the manufacturer's instructions. Isolates were stored in 20% (v/v) glycerol at −80 °C.

## Phage susceptibility testing via spot test

Double agar overlay assay was used to determine the susceptibility of isolates to phages. Briefly, phage stocks were serially diluted in SM buffer containing 50 mM Tris-Cl pH 7.5 (Axil Scientific, Singapore), 100 mM NaCl (Sigma Aldrich, MO, United States), 8 mM $MgCl_2 \cdot 7H_2O$ (VWR), and spotted on bacterial lawn. Agar plates were left to incubate overnight at 35 °C.

## Microtiter MIC assays

Overnight bacterial cultures were adjusted to logarithmic growth phase (until $OD_{630} \sim 0.08$) and further diluted in fresh Mueller Hinton II Broth (Cation-Adjusted) (Becton-Dickinson [BD] BBL™, NJ, United States) to obtain an approximate bacterial inoculum of $10^{5.7}$ colony forming units (CFU)/mL. Serial dilutions of phage were then added to the bacterial culture to achieve different multiplicities of infection. Bacteria-phage mixture was incubated for 20 min at 35 °C. A 100 μL of this mixture was added to each well of the customized 96-well microbroth dilution panels (TREK Diagnostics, OH, United States) lined with antibiotics, in accordance with the manufacturer's recommendations and incubated for 16–20 h at 35 °C. Susceptibility and breakpoints were interpreted according to Clinical & Laboratory Standards Institute[36]. *P. aeruginosa* ATCC 27853 was used as the quality control strain. MIC value is the lowest concentration of an antibiotic at which bacterial growth is completely inhibited, as indicated by clear (non-cloudy) well.

## Time-kill assay

Overnight bacterial cultures were adjusted to logarithmic growth phase (until $OD_{630} \sim 0.08$) and further diluted in fresh Mueller Hinton II Broth (Cation-Adjusted) (Becton-Dickinson [BD] BBL™, NJ, United States) to obtain an approximate bacterial inoculum of $10^6$ CFU/mL. Bacteria were either infected with phage(s) at a multiplicity-of-infection of 100 or added with antibiotics at their achievable

unbound body concentrations. Antibiotics added are aztreonam (0.5 mg/L), ceftazidime/avibactam (0.5/4 mg/L), piperacillin/tazo-bactam (4/4 mg/L), fosfomycin (16 mg/L), levofloxacin (0.5 mg/L), ceftolozane/tazobactam (0.5/4 mg/L) and meropenem (0.5 mg/L). Each antibiotic was used at its MIC level as determined in the microtiter MIC study with the four-bacteria mixture and the three-phage cocktail (Fig. 3C). For samples treated with phage-antibiotics combinations, bacteria were first infected with phage and incubated at 35 °C for 20 min before exposing to antibiotics. All samples were incubated for 24 h at 35 °C in a shaking incubator. At different time points (2, 4, 6, 8 and 24 h), a small aliquot (1 mL) was sampled and plated on Müeller Hinton agar plates for colony enumeration. Synergism between 2 treatment modalities is defined as at least 2 $log_{10}$ CFU per ml reduction from the most active single treatment modality at 24 h[37–39]. Additivity is defined as 1 to less than 2 $log_{10}$ CFU per ml reduction from the most active single treatment modality at 24 h. Bactericidal activity is exhibited as 3 $log_{10}$ CFU per ml reduction from the baseline (at time 0 h) inoculum. Neutrality is defined as less than 1 $log_{10}$ CFU per ml reduction from the most active single treatment modality at 24 h. Antagonism is defined as at least 2 $log_{10}$ increase in CFU/mL at 24 h for the combination compared to the most active single modality.

## Biofilm dispersion assay

We tested the ability of our phage cocktails to disperse biofilm in vitro using crystal violet (CV) staining as a proxy for biofilm biomass, and CFU enumeration as a measure of bacteria viability. Our crystal violet staining protocol is derived from the original work of Christensen et al.[40]. Briefly, *P. aeruginosa* (patient's isolates) were pre-cultured to exponentially growing phase in LB. 1 mL of each culture was pelleted at 8000 rpm for 3 min, washed twice in M63 media and resuspended to a $OD_{600}$ of 0.6. This resuspension was used to inoculate 96-well plate (190 μL of M63 + 10 μL of resuspension). Plates were incubated for 6 h or 24 h, and biofilm was allowed to form on the well's surface before being washed carefully with sodium-magnesium (SM) buffer (50 mM Tris-HCl [pH 7.5], 100 mM NaCl, 10 mM $MgSO_4$) before phage cocktails in SM buffer were added for an additional 24 h. For CV staining and biofilm biomass quantification, supernatant was carefully discarded after 24 h of treatment, wells were washed thrice with MiliQ water, and plates allowed to dry for 1 h. Next, 200 μL of 0.1% crystal violet solution (5% methanol, 5% isopropanol) in MiliQ water was added and plates were rocked for 30 min before rinsing three times and dried for 2 h at room temperature. Finally, 200 μL 20% acetic acid in MiliQ water was added to solubilize the CV before $OD_{595}$ was measured using a plate reader. To determine viability of bacteria within the biofilm, the supernatant was carefully discarded after 24 h of treatment, wells were washed thrice with M63 media and the remaining biofilm was resuspended in LB. Serial dilutions were plated on LB plates and incubated for 24 h before CFU enumeration.

## Phage preparation

Phages PW21, KSY1a and P0413 were generated and purified according to Pirnay et al.[22], with minor modifications. Briefly, phages were co-cultured with bacteria host strain PA2111 in 4 L BBL™ Müeller Hinton II Broth, Cation adjusted, at 35 °C for 16 h. Lysate was filtered through 0.22 μm filter to sterilize, pelleted by centrifugation at 10,000 x g for 90 min at 4 °C and resuspended with Infusol® NS 0.9% (w/v) saline. Endotoxins were depleted by passing the sample through Lionex EndoTrap® HD 1 mL column three times. Endotoxin purified phage suspensions were filtered using medical grade 0.22 μm filter, collected and aliquoted into sterile, depyrogenated 2.0 mL glass vials.

Sterility and bacterial endotoxins testing were performed, according to USP71 and USP85 standards respectively, by Setsco Services Pte Ltd (Singapore). A phage product meets the endotoxin safety threshold for intravenous use if it can be administered at <5 EU/kg/h.

**Phage genomic DNA extraction and whole genome sequencing**

Genomic DNA from pure phage preparation was extracted using Norgen Phage DNA Isolation Kit (Norgen Biotek, Thorold, ON, Canada) according to the manufacturer's guidelines. DNA concentration was measured using the Qubit fluorometer (Thermo Fisher Scientific, Waltham, MA, USA). Raw reads, from NovoSeq systems (Illumina Inc., CA, USA), were trimmed and assembled with an in-house pipeline using Trim Galore (0.6.10) and SPAdes (3.15.5), respectively. Host contamination was identified with bwa-mem2 and removed with in-house bash code using bwa-mem2 and reassembled with SPAdes again. Assembled contigs were then annotated with pharokke (1.7.0). Taxonomy was identified according to the closest match in database with KmerFinder and BLASTn (v2.13.0) against nr. Phage lifestyle was then predicted with PhageAI and the presence/absence of integrase. Assembled contigs of phages were analyzed with ResFinder and CARD RGI to identify AMR genes and were BLASTn against the VFDB (2022-Sep-23 update) to confirm the absence of virulence genes.

**Determination of Phage-driven antibiotic sensitization**

Briefly, growth inhibition assays were carried out in 96-well plates in presence of increasing concentration of either levofloxacin or piper-acillin/tazobactam, combined with either phages alone or combined in a cocktail. OD600 was measured after 24 h of incubation and reported as a function of antibiotic concentration. MIC shift was inferred from these results. Phage-driven antibiotic killing at sub-MIC concentration was determined by treating an inoculum of $10^6$ bacteria with either a sub-MIC concentration of levofloxacin or piperacillin/tazobactam, alone or in combination with phages. Viability was determined after 24 h incubation by CFU enumeration.

**Amplification of Phage DNA from Patient Blood Samples**

Described in Supplementary Information

**Detailed methods and results** for TDM, Synergy E-test, antibiotics susceptibility profile of the *P. aeruginosa* isolates, transmission electron microscopy imaging of the phages, and phage immune monitoring are available in Supplementary Information.

**Statistics & Reproducibility**

No statistical method was used to predetermine sample size. No data were excluded from the analyses. The experiments were not randomized; The investigators were not blinded to allocation during experiments and outcome assessment. Line graphs and scatter plots were generated using GraphPad Prism version 10.0.0 for Windows, GraphPad Software, Boston, Massachusetts USA.

**Reporting summary**

Further information on research design is available in the Nature Portfolio Reporting Summary linked to this article.

## Data availability

All data that support the findings of this study are provided in the article and Supplementary Information/Source Data files. WGS raw reads have been deposited in the NCBI Sequence Read Archive (SRA) under accession numbers listed in the Supplementary Information. Source data are provided with this paper.

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

## Acknowledgements

We would like to thank the department of infectious diseases, department of microbiology, the antimicrobial stewardship team and the nursing team for the clinical support rendered to the phage team and the patient. This publication was supported in parts by SingHealth Group Allied Health/SingHealth Group Pharmacy. This work was supported by Singapore Ministry of Health National Medical Research Council NMRC IRG (MOH-000957, A.L-H. K.), NMRC SMART III (CG21APR1011, A.L-H. K.), NMRC CoSTAR-HS (CG21APR2005, A.L-H. K.), Clinical Scientist Awards (MOH-001293-01, MOH-001278-01, MOH-001168-00 and MOH-000018-00, A.L-H. K.), NMRC– Open Fund Young Individual Research Grant (MOH-OFYIRG25jan-0007, T.T.A.), SingHealth Duke-NUS Academic Medicine Philanthropic Funds & Singapore General Hospital Academic Medicine General Fund (AMSGH/03-09/FY2024/EX/55-A116(a), A.L-H. K.), Open Fund Individual Research Grant (MOH-OFIRG21jun0038, W.M.). We also acknowledged financial support from institutional grants: Singapore General Hospital Research Grant (SRG-OPN-02-2024, S.T.) and (SRG-OPN-03-2025, S.T.), and from the Ministry of Education, Singapore, under its Research Center of Excellence award to the Institute for Digital Molecular Analytics & Science, NTU (IDMxS, grant: EDUNC-33-18-279-V12, W.M.) and the Singapore Center for Environmental Life Sciences Engineering, NTU (SCELSE, grant: EDUN C33-62-036-V4, W.M.).

## Author contributions

A.L-H. K., W.M., and S.J.C. designed and supervised the work. S.J.C., Y.L., W.M., and A.L-H.K. were involved in the conception and writing of the manuscript. S.T., T.T.A., A.L., S.T., Z. L. M.G.K., W.M., and A.L-H.K. were part of the Singapore Phage Repository group and provided therapeutic phages. S.J.C., D.H.L.N., N.G.S.C., W.H.L., T.P.L., and A.L.K., L.W., T.Y.C., T.H.W.N., L.Y.J., T.T.E., and T.T.T. provided the relevant clinical data and perspectives, while Y.L., S.T., Z.S.C., J.H.Y., Y.Z., Z.L., S.T., M.G.K., B.H.T., P.H.E.Y., T.T.A., R.T.O., K.K.K.K., T.H.N.W., W.M., and A.L-H.K. contributed microbiological and laboratory data to the laboratory workup, analysis and interpretation of laboratory data. S.T., Z.S.C., Z.L., S.T., S.K., and J.H.Y. performed all phage related experimental work. K.K.K.K., M.G., S.K., and Y.Z. performed WGS and bioinformatic analyses. W.M. provided data on biofilm assays and antibiotic-sensitization mechanism. All authors provided critical feedback and helped shape the study, analysis, and manuscript.

## Competing interests

The authors declare no conflict of interest.

## Additional information

Shimin Jasmine Chung [1,2,3,19], Yang Liu [4,5,19], Shuhua Thong[4,6], Yang Zhong[4,7], Zhi Soon Chong [4], Zhining Lim[8,9], Sabrina Tan[8,9], Jia Hao Yeo [4,6], Ming Guang Koh[8,9], Nathalie Grace Sy Chua [4], Dorothy Hui Lin Ng[1,3], Winnie Hui-Ling Lee[4], Tze Peng Lim[4,6], Limin Wijaya[1,3], Boon Huan Tan[10], Peng Huat Eric Yap[11], Thet Tun Aung[12,13], Rick Twee-Hee Ong[14], Karrie Kwan Ki Ko [6,15], Tse Hua Nicholas Wong[6,15], Yu Lin Charlene Tang[16], Yee Jim Loh[17], Teing Ee Tan[2,17], Thuan Tong Tan[1,2,3], Sandra Kolundzija[8,9], Wilfried Moreira[8,9,20] ✉ & Andrea Lay-Hoon Kwa [3,4,18,20] ✉

[1]Department of Infectious Diseases, Singapore General Hospital, Singapore, Singapore. [2]SingHealth Duke-NUS Transplant Centre, Singapore, Singapore. [3]SingHealth-Duke-National University of Singapore Medical School, Academic Clinical Programme (Medicine), Singapore, Singapore. [4]Division of Pharmacy, Singapore General Hospital, Singapore, Singapore. [5]PhD Clinical and Translational Sciences, Duke-National University of Singapore Medical School, Singapore, Singapore. [6]SingHealth-Duke-National University of Singapore Medical School, Academic Clinical Programme (Pathology), Singapore, Singapore. [7]Department of Clinical Translational Sciences, Singapore General Hospital, Singapore, Singapore. [8]Singapore Centre for Environmental Life Science Engineering (SCELSE), Nanyang Technological University (NTU), Singapore, Singapore. [9]Institute for Digital Molecular Analytics and Science (IDMxS), Nanyang Technological University (NTU), Singapore, Singapore. [10]Research Laboratory Division, Communicable Diseases Agency, Singapore, Singapore. [11]Lee Kong Chian School of Medicine, Nanyang Technological University, Singapore, Singapore. [12]Ocular Infections and Anti-Microbials Research Group, Singapore Eye Research Institute, Singapore, Singapore. [13]Ophthalmology and Visual Sciences Academic Clinical Program, Duke-NUS Medical School, Singapore, Singapore. [14]Saw Swee Hock School of Public Health, National University of Singapore, Singapore, Singapore. [15]Department of Microbiology, Singapore General Hospital, Singapore, Singapore. [16]Department of Nuclear Medicine and Molecular Imaging, Singapore General Hospital, Singapore, Singapore. [17]Department of Cardiothoracic Surgery, National Heart Centre, Singapore, Singapore. [18]Emerging Infection Diseases Program, Duke-National University of Singapore Medical School, Singapore, Singapore. [19]These authors contributed equally: Shimin Jasmine Chung, Yang Liu. [20]These authors jointly supervised this work: Wilfried Moreira, Andrea Lay-Hoon Kwa ✉e-mail: wilfried.moreira@ntu.edu.sg; andrea.kwa.l.h@sgh.com.sg

