## [Transparent Peer Review file · Nature Communications]

Timely bespoke bacteriophage-antibiotic combination successfully treats refractory *Pseudomonas aeruginosa* mediastinitis and vascular graft infection

Corresponding Author: Dr Andrea Kwa

Version 0:

Reviewer comments:

Reviewer #1

(Remarks to the Author)

This case report describes the use of personalized phage and antibiotic combination therapy to manage a refractory *Pseudomonas aeruginosa* vascular graft infection. The patient experienced multiple episodes of recurrent bacteremia despite extended courses of high dose antibiotics, including piperacillin tazobactam and fluoroquinolones. Phage therapy was pursued due to the persistent biofilm nature of the infection, the emergence of fluoroquinolone resistance, and the elevated surgical risk that ruled out graft removal.

A three-phage cocktail was selected based on its *in vitro* lytic activity and biofilm disruption capacity. It included two closely related Pbinavirus phages (PW21 and P0413) and a jumbo PhiKzvirus phage (KSY1a). Phage therapy was administered IV for two weeks alongside ongoing treatment with piperacillin tazobactam and levofloxacin. At the time of phage initiation, blood cultures were negative, indicating resolution of bacteremia, but 18F FDG PET/CT imaging continued to show metabolically active infection.

The addition of phage therapy was considered relatively safe. The patient experienced no clinical adverse events and remained well (line 281), despite a transient but measurable rise in inflammatory markers including CRP and ESR. Although these returned to baseline after treatment (lines 283 to 289). Imaging on October 24 showed radiologic improvement, and the patient was transitioned to oral fluoroquinolone suppression therapy.

The study contains several strengths, including a multidisciplinary treatment approach, early integration of phage therapy into the care plan, rigorous screening of bacterial isolates, and therapeutic drug monitoring. However, it also has important limitations. The main concern is that the evidence supporting phage efficacy is circumstantial and mechanistically plausible but not definitive. The patient was already showing improvement on the September 10 PET/CT scan, prior to the initiation of phage therapy on September 23. The next scan was not performed until October 24, which was 18 days after the end of phage treatment. This delay makes it impossible to attribute radiologic improvement specifically to phages, as the observed effect may also reflect ongoing antibiotic therapy, immune responses, or natural recovery.

Clinically, the infection was not cured but managed through suppression, as shown by the patient's continued need for oral fluoroquinolone therapy. Although the case demonstrates that phage therapy was safe and feasibly implemented, the lack of post-treatment microbiological biomarkers or mechanistic data prevents any firm conclusion that phages contributed to clinical improvement. This evidentiary gap is common across most compassionate use phage therapy reports.

The *in vitro* data showed reduced MICs and enhanced bacterial killing in the presence of phages, but the study did not use standard synergy testing methods. It lacked checkerboard assays and/or time kill studies with set thresholds compared to the most active single agent. In addition, the phage antibiotic combinations were not tested against appropriate controls. These omissions limit the strength of the claim that phage and antibiotic synergy drive clinical improvement.

The increase in inflammatory markers following phage therapy is described as mild and expected. However, from a critical standpoint, the elevations in CRP and ESR, though transient, may represent a meaningful systemic immune response to intravenous phage administration, even in a clinically stable patient. This possibility was not explored in detail and deserves further investigation, especially in relation to immunogenicity and repeat dosing.

The assertion that phage therapy restored fluoroquinolone susceptibility is also not well supported. While post-treatment MICs were lower and the patient tolerated oral levofloxacin, the study did not include sequencing or resistance profiling of post-treatment isolates. Therefore, it is unclear whether the susceptible strain was a direct descendant of the fluoroquinolone resistant strains identified earlier. The change in susceptibility could reflect strain replacement, selection of a preexisting susceptible subpopulation, or natural variation rather than true reversion of resistance.

Overall, the innovation in this study lies in the operational and clinical framework for delivering phage therapy within a modern infectious disease care model. That contribution is valuable and relevant for translation into real world practice. However, the mechanistic and pharmacodynamic claims regarding phage efficacy should be interpreted cautiously, given the limitations in microbiological evidence, timing, and experimental rigor.

Minor comments.

While the study uses standard translational phage microbiology tools (spot testing, time-kill assays, biofilm assays), it is unclear whether these assays led to a meaningfully more effective treatment strategy than other possible phage cocktail options in vivo. The authors should clarify how these in vitro results translated into the final clinical decision.

Line 9: Proofreading is needed.

Line 47: Replace “well-planned and well-timed” with “systematically planned and timely” to avoid redundancy.

Lines 60–61: The sentence describing post-phage therapy improvement is redundant (“reduced tracer uptake... no recurrence of bacteremia” is repeated). Recommend condensing.

Line 89: Change “bacterial eradication rate was achieved...” to “bacterial eradication was achieved in 6 of 7 cases...” for clarity.

Lines 95–98: The description of the patient’s complex surgical history could benefit from minor simplification or expanded detail to improve readability and clinical context.

Lines 283–284: The phrase “slight increase in inflammatory markers” downplays the quantitative data (CRP 16.8 mg/L, ESR 44 mm/h). Consider rephrasing as a “measurable increase” to reflect biological significance.

Line 303: The report of “100% neutralization with sharp recovery” for PW21 is biologically implausible. Clarify whether this result is due to technical variability, assay noise, or another artifact.

Lines 314–319: The sentence describing the phage therapy framework is overly long and contains multiple clauses. Recommend splitting into two sentences.

Line 342: Rephrase “this is the index phage therapy case in Singapore” as “this represents the first documented case of phage therapy in Singapore”, especially in light of the surrounding discussion and critique of causality.

Lines 367–375: Suggest including a brief reference to standard definitions of synergy.

Lines 400–403: The discussion of cost savings is compelling. Consider adding a concluding sentence framing this as a potential health system benefit, especially in resource-limited or outpatient settings.

Figure 2 – PET/CT Imaging. Add scale bars for spatial reference. Use consistent phrasing in captions to label timepoints clearly (e.g., “Pre-phage” [July], “Post-phage” [October]) to guide readers through the treatment timeline.

Figure 3 – Define “LOD” (Limit of Detection).

Figure 4 – Add caption labels to key steps in the workflow (e.g., TDM, iACT, Phage Screening). Visually distinguish parallel vs. sequential processes to clarify the integrated treatment approach.

Reviewer #2

(Remarks to the Author)

This is a nice exposition of a clinical case with a successful outcome. Initial failures are described by the authors as potentially attributable to inadequate antibiotic therapy. It seems that success followed further surgical debridement and improved antibiotic management, as well as a course of bacteriophage therapy.

They also describe four members of what we presume to be a diversified population of *P. aeruginosa*, including a shift in susceptibility toward quinolones.

Some of the assays are slightly unconventional (mixtures of bacteria, high phage MOIs etc) and the authors show that the phages were not highly active, even in combination, under the in vitro conditions tested. They show antibiotic synergy, which is interesting as it includes a quinolone and one (but not most) beta-lactams.

It would be ideal to show that the phages themselves contributed to outcome, or at least that some sort of amplification of phage numbers were detected as a surrogate for successful predation. The samples required to show this are not always available, of course.

In order to enhance the manuscript, I have a number of more specific suggestions:

1. The differences between the *P. aeruginosa* isolates are not very clear to me at least, as only % identity, an ST, an antibiogram and a gene list are provided but these are presumably four members of a diversified population of bacteria under stress. The CAZ-AVI and CIP synergy are interesting, as neither ab alone provides sustained growth suppression, nor do any of the phage combinations. This would be worth a little more explanation, as it bears significantly on differential phage responsiveness. Relevant aspects to discuss are bacterial defence systems and prophages in particular I think.
2. It would also be interesting to discuss the LEVO susceptibility shift in kinetics and contrast the mechanism of the CAZ AVI effect, since it has such a different mode of action. Do the bacterial genomic data explain this in any way?
3. I think the supp data on phage-antibiotic synergy might be more strongly featured and discussed, including justification for the slightly unusual choice to combine all 4 bacteria as a mixture in testing, and to use the very high MOI of 100. Are there

any data at subMIC in support of synergy?

4. Fig S9 shows no real suppression of growth by any phage combination beyond 6-8h, apparently using a very high MOI of 100. Are there any data at lower MOIs?
5. Sharp initial collapses (2-3 log) in control growth kinetics are not discussed in Figs S8-14. Could the authors please comment on this and its impact on the readers interpretation of the time-kill curves.
6. I saw no data regarding the presence of phages in blood – do we know what clearance kinetics were like?
7. Do we have any evidence of phage amplification to show that phages were successfully preying on the bacterial population/s?
8. Was there any evidence of a host inflammatory reaction to the phage administration?
9. Were any (targeted) bacteria or (administered) phages recovered from wounds/ mediastinum/ tissue or body fluids from which we might learn about any co-adaptation? What was done to retrieve them (eg it may be that there were no pseudomonas and no phages present soon after initial phage administration) ?
10. The serum neutralization data are a little confusing as well – it seems that the serum was diluted to low levels (1%) and the bacterial density was high (in blood, a 10^9 dose will be diluted immediately to 10^4 - 10^5 etc) – it might be better to set these data aside completely or to add in more data/ revise that section significantly to make it more impactful/ informative.
11. With regard to layout/ content:
 - a. Fig 4 is unnecessary I think, or could move to / combine with Fig S2
 - b. Fig 3 could be simplified by omitting EMs and spot tests, and referring instead to Fig S3
 - c. Overall, the text could be greatly reduced and the focus shifted to those questions that are raised – eg is it the case that phages which are apparently poor performers in vitro can be highly effective in vivo? And to highlight the evidence for this eg phage amplification, host responses, bacterial lysis products in blood etc.

Reviewer #3

(Remarks to the Author)
Dear Author,

In this Case Report to the Journal, you report on a young female patient with an early refractory thoracic vascular graft infection following aortic root replacement and pulmonary homograft replacement that was successfully treated with a targeted combination of phages and antimicrobial treatment. This report highlights the novel use of phage therapy in a case where a re-operation with complete removal of the infected prosthesis was deemed not possible due to a history of several re-sternotomies (5 in total) and a molesting biofilm-producing bacteria (*Pseudomonas aeruginosa*). After achieving multidisciplinary consensus and following several bacteremia episodes under antimicrobial treatment, the authors opted to rapidly screen the patient for potential phage therapy and therapeutic drug monitoring. She was successfully treated with phages for two weeks, achieving inflammation control without the need for further intravenous antimicrobial treatment. At follow-up, the patient remains on suppressive oral treatment.

The document is well written and the analysis as well as the conclusion are sound.

From a clinical point of view, there are several questions to be answered, before accepting this interesting manuscript for publication.

Line 106: Why was mediastinal lavage performed 6 days after diagnosis of mediastinitis, why not earlier? And what did you perform exactly?

- Was a tissue flap (muscle/omental) coverage of the prosthesis considered? The group of Hannover showed that this approach was feasible when treating early thoracic vascular graft infections (Umminger et al, Eur Jour Cardiothorac Surg, 2016)
- What did you find intraoperatively, pus? Were intraoperative samples sent for culture? And if so, did they grow positive for the same microorganism?

Line 185: I agree with you that this patient was not operable due to the number of re-sternotomies and the acuity of the infection and the last surgery. However, did you consider at any point to do a re-sternotomy with mediastinal-irrigation and flap-coverage instead of re-operation on cardiopulmonary bypass and explantation of the infected prosthesis?

Line 195: “Definitive surgery to remove the composite vascular graft was not possible”. Again, I agree with you that the patient is not operable, but the wording might not be precise enough, since it is always possible.

When and how was the last follow-up performed? Did you perform an FDG-PET/ CT to control tracer uptake and steer antimicrobial treatment?

Best regards,

Version 1:

Reviewer comments:

Reviewer #1

(Remarks to the Author)

The authors have submitted a thoughtful and comprehensive revision that directly addresses the concerns raised in my initial review. The revised manuscript now presents a well-organized, clearly written, and mechanistically grounded account of a complex *Pseudomonas aeruginosa* vascular graft infection managed with a personalized phage–antibiotic combination. The addition of in vitro time-kill synergy assays, biofilm dispersion experiments, and mutant data demonstrating the role of the MexAB–OprM efflux pump as a phage receptor, and more, meaningfully establish mechanistic plausibility for antibiotic re-sensitization and demonstrate that the phage component of therapy had a rational biological basis rather than being empirically applied.

The authors have also improved clarity throughout the manuscript. The narrative flow between clinical management, microbiological work-up, and mechanistic interpretation is coherent and easy to follow. The revised figures, particularly those illustrating the phage therapy workflow and the integrated clinical–laboratory process, are well designed and enhance readability. Importantly, the authors have moderated earlier overstatements about causality and now appropriately describe the contribution of phage therapy as plausible but lean toward being ultimately circumstantial.

Some limitations inevitably remain with an $n = 1$ study. The assertion that phage therapy “restored” fluoroquinolone susceptibility, while mechanistically supported by the MexAB–OprM data, remains unconfirmed given the absence of post-treatment isolates. It may also be helpful to clarify that the detected phage DNA indicates presence rather than demonstrated replication. The discussion could be slightly condensed to maintain focus.

Reviewer #3

(Remarks to the Author)

Dear authors,

Thank you for addressing these points. I have no further comments.

“Timely bespoke bacteriophage–antibiotic combination successfully treats refractory Pseudomonas aeruginosa mediastinitis and vascular graft infection.”

Point-by-Point Response to Reviewers Comments

Reviewer #1

We thank the reviewer for their positive assessment of the manuscript, particularly noting the strengths of the multidisciplinary approach, early integration of phage therapy, rigorous bacterial isolate screening, and therapeutic drug monitoring. We also appreciate the recognition that the innovation in this study lies in the operational and clinical framework for delivering phage therapy within a modern infectious disease care model, which we agree is valuable and relevant for translation into real-world practice. We acknowledge the concerns regarding the circumstantial nature of the evidence for phage efficacy. Below, we provide a detailed, point-by-point response to each specific comment raised, including clarifications, additional analyses, and manuscript revisions to address these limitations.

[Point #1]

The patient was already showing improvement on the September 10 PET/CT scan, prior to the initiation of phage therapy on September 23. The next scan was not performed until October 24, which was 18 days after the end of phage treatment. This delay makes it impossible to attribute radiologic improvement specifically to phages, as the observed effect may also reflect ongoing antibiotic therapy, immune responses, or natural recovery.

We thank the reviewer for this important observation regarding the timing of imaging and its impact on causal inference. We agree that the evidence is circumstantial and that radiologic improvement cannot be attributed to phage therapy with absolute certainty. However, PET/CT findings provide additional context that supports a likely beneficial contribution of phage therapy.

Specifically, despite prolonged high-dose antibiotic therapy since April 2024, tracer uptake around the aortic graft persisted. Between May 2024 (Fig. 2C) and July 2024 (Fig. 2D), the modest reduction in uptake was temporally associated with antibiotic dose escalation in June 2024. From July 2024 to September 2024 (Fig. 2E), tracer uptake around the graft increased despite continued high-dose piperacillin–tazobactam and levofloxacin, suggesting limited efficacy of antibiotics alone against this likely biofilm-driven process. Reduction in tracer uptake around the graft was only observed following initiation of phage therapy in September 2024, as documented in the October 2024 PET/CT scan.

We have revised Figure 2 to include an updated April 2025 PET/CT demonstrating sustained improvement on levofloxacin monotherapy (despite only intermediate susceptibility of *P. aeruginosa* to fluoroquinolones). The nuclear medicine interpretation, corroborated by our co-author and clinical specialist who reviewed the serial scans, supports a likely contributory role of phage therapy in the observed radiological improvement. The figure legend has been updated to reflect this interpretation and temporal sequence.

“Timely bespoke bacteriophage–antibiotic combination successfully treats refractory Pseudomonas aeruginosa mediastinitis and vascular graft infection.”

Although we cannot definitively attribute the patient’s improvement to phage therapy alone, its use was a deliberate, multidisciplinary decision aimed at addressing the ongoing biofilm-associated infection that had not resolved with antibiotics. The patient's subsequent clinical stabilisation and radiologic improvement, in conjunction with in vitro synergy data and therapeutic monitoring, suggest a likely contributory role of phage therapy to the observed outcome.

Modifications to the manuscript:

- Figure 2, page 9
- Legend of Figure 2, page 10

[Point #2]

Clinically, the infection was not cured but managed through suppression, as shown by the patient’s continued need for oral fluoroquinolone therapy. Although the case demonstrates that phage therapy was safe and feasibly implemented, the lack of post-treatment microbiological biomarkers or mechanistic data prevents any firm conclusion that phages contributed to clinical improvement. This evidentiary gap is common across most compassionate use phage therapy reports.

We thank the reviewer for this comment and appreciate the opportunity to provide additional clinical context. Despite receiving appropriate culture-directed antibiotic therapy in April–May 2024, the patient experienced breakthrough *P. aeruginosa* bacteremias, documented on two occasions through frequent surveillance blood cultures (without waiting for clinical symptoms manifestation) performed due to the complexity of her case. Clinical symptoms were minimal, and inflammatory markers remained within normal limits even during these episodes of confirmed bacteremia, underscoring the muted biochemical and clinical responses that complicated assessment of treatment efficacy (Fig. 1).

Serial PET/CT imaging, however, demonstrated increasing tracer uptake around the aortic valve prosthesis despite high-dose antibiotics, consistent with persistent deep-seated infection. Following initiation of phage therapy, we observed (1) radiological improvement on PET/CT, and (2) no recurrence of bacteremia on surveillance cultures through April 2025, despite the patient being maintained on levofloxacin monotherapy since December 2024 (with *P. aeruginosa* isolates of intermediate fluoroquinolone susceptibility). We have updated Figure 2 to reflect this longitudinal imaging.

In cases such as this—vascular graft infections with recurrent bacteremia not amenable to surgical replacement—long-term suppressive antibiotic therapy is standard practice and does not necessarily represent failure of phage therapy. The decision to continue suppression is made on an individual basis through shared decision-making with the patient, given the high risk associated with treatment withdrawal.

“Timely bespoke bacteriophage–antibiotic combination successfully treats refractory Pseudomonas aeruginosa mediastinitis and vascular graft infection.”

Finally, while we lack post-treatment microbiologic biomarkers, we note and agree that this constraint is typical of compassionate-use protocols, where invasive sampling is clinically contra-indicated. Similarly, while mechanistic confirmation of phage activity *in vivo* is commonly difficult to demonstrate, we provided additional *in vitro* data that establish a mechanistic basis for the synergistic effect of the phage and antibiotic combination (see next point).

Modifications to the manuscript:

- We have revised Figure 1, and the Legend of Figure 1 on pages 7 and 8, respectively for clarity.

[Point #3]

The in vitro data showed reduced MICs and enhanced bacterial killing in the presence of phages, but the study did not use standard synergy testing methods. It lacked checkerboard assays and/or time kill studies with set thresholds compared to the most active single agent. In addition, the phage antibiotic combinations were not tested against appropriate controls. These omissions limit the strength of the claim that phage and antibiotic synergy drive clinical improvement.

We thank the reviewer for this important observation. We agree that the initial synergy testing lacked the full range of standardized methodologies. In response to this feedback, we have now conducted additional experiments incorporating time-kill curves using phage alone, antibiotic alone, and their combinations, with clearly defined synergy thresholds ($\geq 2 \log_{10}$ increase in killing at sub-MIC concentration). These new data have been included in the revised manuscript (Figure 4) and demonstrate reproducible synergy between the phage cocktails and key antibiotics against the patient isolates. Further to this, we tested our in-house *P. aeruginosa* transposon mutants *Tn::mexB*, deficient for the MexAB-OprM membrane transporter complex, and showed that it constitutes a key determinant of efficacy for several of our phages *i.e.* the receptor for these phages. MexAB-OprM is an efflux transporter that has been shown to confer resistance to penicillins and quinolones. We (and others - see references below) have shown that the disruption of MexAB-OprM leads to sensitization to these antibiotics. We believe these additional findings strengthens the mechanistic foundation for the observed clinical outcome and enhances the translational value of our findings. We have incorporated these findings in Figure 3 and Supplementary Figures.

References:

- Chan, B., Sstrom, M., Wertz, J. et al. Phage selection restores antibiotic sensitivity in MDR *Pseudomonas aeruginosa*. *Sci Rep* 6, 26717 (2016). <https://doi.org/10.1038/srep26717>
- Ho P, Dam LC, Koh WRR, Nai RS, Nah QH, Rajaie Fizla FBM, Chan CC, Aung TT, Goh SG, Fang Y, Lim Z, Koh MG, Demott M, Boucher YF, Malleret B, Gin KY, Dedon P, Moreira W. Screening of the PA14NR Transposon Mutant Library

“Timely bespoke bacteriophage–antibiotic combination successfully treats refractory Pseudomonas aeruginosa mediastinitis and vascular graft infection.”

Identifies Genes Involved in Resistance to Bacteriophage Infection in Pseudomonas aeruginosa. Int J Mol Sci. 2024 Jun 26;25(13):7009. doi: 10.3390/ijms25137009.

Modifications to the manuscript:

- We created a Figure 3 (Phage Therapy Workflow) on page 13, and revised Figure 4 (which was previously figure 3) and its legend on page 16 and 17, respectively
- Figure 4 describes the mechanisms behind antibiotic-sensitization, MIC shift and increased killing at sub-MIC of antibiotics.
- Additional Supplementary Figure S17 showing the role of MexAB-OprM in phage efficacy
- Results section, pages 14-17, lines now 363-462.
- Discussion section, page 19-20, lines 531-558.

[Point #4]

The increase in inflammatory markers following phage therapy is described as mild and expected. However, from a critical standpoint, the elevations in CRP and ESR, though transient, may represent a meaningful systemic immune response to intravenous phage administration, even in a clinically stable patient. This possibility was not explored in detail and deserves further investigation, especially in relation to immunogenicity and repeat dosing.

We agree that the transient rise in CRP and ESR could reflect a systemic immune response to intravenous phage therapy. However, the patient remained clinically stable and asymptomatic throughout the course of Phage Therapy, and no clinical event correlated with the CRP and ESR trends. Nonetheless, we have now expanded the discussion to address potential immunogenicity and the need for additional study in future cases, particularly in the context of repeat dosing and immunological markers.

Modification to manuscript:

- Discussion section, page 20, lines 560-562.

[Point #5]

The assertion that phage therapy restored fluoroquinolone susceptibility is also not well supported. While post-treatment MICs were lower and the patient tolerated oral levofloxacin, the study did not include sequencing or resistance profiling of post-treatment isolates. Therefore, it is unclear whether the susceptible strain was a direct descendant of the fluoroquinolone resistant strains identified earlier. The change in susceptibility could reflect strain replacement, selection of a preexisting susceptible subpopulation, or natural variation rather than true reversion of resistance.

We thank the reviewer for highlighting this important point. Considering this and as discussed above, we used a *P. aeruginosa* MexAB-OprM receptor-deficient mutant and tested its susceptibility to our phage cocktail. The mexAB-OprM complex has been

“Timely bespoke bacteriophage–antibiotic combination successfully treats refractory Pseudomonas aeruginosa mediastinitis and vascular graft infection.”

previously been identified (see Point #3) as a major receptor for *P. aeruginosa* phages. Its deletion leads to increased sensitivity to Fluoroquinolones and Beta-lactams. Phages targeting the MexAB-OprM complex have been used both in vitro and in vivo and achieved phenotypic trade-off between antibiotic resistance and phage sensitivity. We tested the ability of the phages used in our therapeutic cocktail to infect this mutant. The results showed that 1 out of 3 of the phages composing our therapeutic cocktail rely on MexAB-OprM to carry out a lytic infection. These findings provide a mechanistic rationale for the antibiotic sensitization observed *in vitro*, and likely *in vivo*. We have revised the manuscript to present these critical findings (Figure 4 and supplementary Figure S17). We also acknowledge the need of post-treatment isolate characterization explicitly and propose genomic follow-up as a critical component for future studies.

Modification to the manuscript:

- Figure 4
- Additional Supplementary Figure S17 showing the role of MexAB-OprM in phage efficacy
- Discussion section, page 20, lines 553-555.

[Point #6]

While the study uses standard translational phage microbiology tools (spot testing, time-kill assays, biofilm assays), it is unclear whether these assays led to a meaningfully more effective treatment strategy than other possible phage cocktail options in vivo. The authors should clarify how these in vitro results translated into the final clinical decision.

We thank the reviewer for this valuable observation. The in vitro assays (particularly the phage/antibiotic combination kill kinetic and biofilm disruption assays) were used to prioritize phages with complementary activity profiles, including rapid lytic kinetics, efficacy against biofilm-embedded cells, and performance in combination with the patient's antibiotics. These results informed the clinical decision to select a cocktail targeting both planktonic and biofilm-associated bacterial populations. However, we acknowledge that in the context of a single compassionate-use case, it is not possible to definitively determine whether this combination was the most effective among all possible phage and antibiotics combinatorial options. We have clarified this uncertainty in the revised manuscript (revised Figure 3 describing our phage therapy workflow) and explicitly linked the *in vitro* findings to the clinical selection process (Figure 4 linking the phage selection process to the clinical treatment decision)

Modification to the manuscript:

- Revised Figure 3 and Legend of Figure 3; Results section, page 13
- Revised Figure 4 and Legend of Figure 4; Results section, page 16 -17 ; Discussion section, page 19 lines 528-530

“Timely bespoke bacteriophage–antibiotic combination successfully treats refractory Pseudomonas aeruginosa mediastinitis and vascular graft infection.”

Minor comments.

Line 9: Proofreading is needed. Thank you. We revised it.

Line 47: Replace “well-planned and well-timed” with “systematically planned and timely” to avoid redundancy. Thank you for the recommendation. The text has been revised. Please see line now 49.

Lines 60–61: The sentence describing post-phage therapy improvement is redundant (“reduced tracer uptake... no recurrence of bacteremia” is repeated). Recommend condensing.

We have revised the sentences to the following: “Post phage therapy, radiological improvements were seen on serial PET/CT and bacteremia was eliminated, without recurrence. The patient was transitioned to an oral fluoroquinolone maintenance treatment” in lines 56-58

Line 89: Change “bacterial eradication rate was achieved...” to “bacterial eradication was achieved in 6 of 7 cases...” for clarity. The statement has been revised as recommended in lines now 102-103.

Lines 95–98: The description of the patient’s complex surgical history could benefit from minor simplification or expanded detail to improve readability and clinical context. To improve clarity, we have revised the patient’s complex surgery as follow in lines now 106-113: “A 36-year-old female patient with a history of congenital heart disease (pulmonary atresia with ventricular septal defect) with 4 prior sternotomies for corrective heart surgery developed progressive dilatation of her ascending aorta with severe aortic regurgitation and pulmonary homograft stenosis in 2023. This was associated with admissions for decompensated heart failure. To address this, she eventually underwent a Bentall procedure where the affected aortic valve, aortic root and ascending aorta were replaced by a composite vascular graft and the pulmonary homograft was also replaced in the same setting on January 25th, 2024, just 5.5 weeks after she had delivered her second child.”

Lines 283–284: The phrase “slight increase in inflammatory markers” downplays the quantitative data (CRP 16.8 mg/L, ESR 44 mm/h). Consider rephrasing as a “measurable increase” to reflect biological significance. The statement has been rephrased in line 312.

Line 303: The report of “100% neutralization with sharp recovery” for PW21 is biologically implausible. Clarify whether this result is due to technical variability, assay noise, or another artifact.

We thank this reviewer for this observation. We believe that these results are due to technical variability or inadequacy and, in accordance with reviewer 2 who has raised a similar point and suggested to leave these results out, we propose to remove these findings in accordance with reviewer 2’s suggestion.

Modifications to the manuscript:

- Removal of Supplementary Figure S16 and its description.
- Removal of the immune neutralisation description of phages before and during phage therapy in previous page 20 of the first submitted manuscript version.
- Discussion section Page 20 now : We have added a comment on the need for immunological studies on phage neutralisation throughout PT in lines 567.

“Timely bespoke bacteriophage–antibiotic combination successfully treats refractory Pseudomonas aeruginosa mediastinitis and vascular graft infection.”

Lines 314–319: The sentence describing the phage therapy framework is overly long and contains multiple clauses. Recommend splitting into two sentences.

We have revised the sentences in lines now 510-530. The revised section is as follows: “Despite months of high-dose, TDM-guided antibiotic therapy with iACT optimisation, serial 18F-FDG PET/CT scans demonstrated persistent tracer uptake on the aortic graft, suggesting likely long-term treatment failure (...). Adjunctive phage therapy was considered early (*i.e.* before deterioration) in April 2024 (...). In this report, we propose a framework for the timely systematic incorporation of compassionate adjunctive phage therapy (...).”

Line 342: Rephrase “this is the index phage therapy case in Singapore” as “this represents the first documented case of phage therapy in Singapore”, especially in light of the surrounding discussion and critique of causality.

We have revised line 342 (now 528-529) to “.....marking Singapore’s first reported clinical phage use”.

Lines 367–375: Suggest including a brief reference to standard definitions of synergy. Basic definition (synergy = >2-log increased killing) with the references below are added to Method section page 24 lines 664-671.

References:

- Comeau AM, Tétart F, Trojet SN, Prère M-F, Krisch HM (2007) Phage-Antibiotic Synergy (PAS): β -Lactam and Quinolone Antibiotics Stimulate Virulent Phage Growth. PLoS ONE 2(8): e799. <https://doi.org/10.1371/journal.pone.0000799>

- Morris TC, Reyneke B, Khan S, Khan W. Phage-antibiotic synergy to combat multidrug resistant strains of Gram-negative ESKAPE pathogens. Sci Rep. 2025 May 18;15(1):17235. doi: 10.1038/s41598-025-01489-y. PMID: 40383795; PMCID: PMC12086229.

- Nicolle P, Faguet M. La synergie lytique de la pénicilline et du bactériophage, étudiée au microbiophotomètre [The lytic synergy of penicillin and bacteriophage, studied with a microbiophotometer]. Ann Inst Pasteur (Paris). 1947 May;73(5):490-5. French. PMID: 20253582.

*Lines 400–403: **The discussion of cost savings is compelling.** Consider adding a concluding sentence framing this as a potential health system benefit, especially in resource-limited or outpatient settings.*

We thank the reviewer for this positive comment. In the section conclusion, this is commented and concluded in lines now 601-604

Figure 2 – PET/CT Imaging. Add scale bars for spatial reference. Use consistent phrasing in captions to label timepoints clearly (e.g., “Pre-phage” [July], “Post-phage” [October]) to guide readers through the treatment timeline.

Figure 2 has been amended to include representative PET-CT images in Apr 2025. The timepoint labels are also revised based on the above recommendations. A scale bar has also been added.

Figure 3 –Define “LOD” (Limit of Detection).

“Timely bespoke bacteriophage–antibiotic combination successfully treats refractory Pseudomonas aeruginosa mediastinitis and vascular graft infection.”

The limit of detection (LOD) in a time kill assay here refers to the lowest bacterial cell concentration that can be reliably detected using plate count method. Definition has been added.

Figure 4 –Add caption labels to key steps in the workflow (e.g., TDM, iACT, Phage Screening). Visually distinguish parallel (color coded) vs. sequential processes (directed by arrows) to clarify the integrated treatment approach.

We have significantly revised Figure 4 (now Figure 5) to reflect the reviewer’s suggestions. Figure 5 now presents 3 tracks (Patient Journey, Clinical Process and Phage Pipeline) which illustrate the integrated treatment approach, with detailed sequential and parallel processes and arrows linking each tracks.

Modification to the manuscript:
- New Figure 5 (ex-Figure 4), page 22

Reviewer #2

[Point #1]

*This is a nice exposition of a clinical case with a successful outcome. Initial failures are described by the authors as potentially attributable to inadequate antibiotic therapy. It seems that success followed further surgical debridement and improved antibiotic management, as well as a course of bacteriophage therapy. They also describe four members of what we presume to be a diversified population of *P aeruginosa*, including a shift in susceptibility toward quinolones. Some of the assays are slightly unconventional (mixtures of bacteria, high phage MOIs etc) and the authors show that the phages were not highly active, even in combination, under the in vitro conditions tested. They show antibiotic synergy, which is interesting as it includes a quinolone and one (but not most) beta-lactams. It would be ideal to show that the phages themselves contributed to outcome, or at least that some sort of amplification of phage numbers were detected as a surrogate for successful predation. The samples required to show this are not always available, of course.*

We thank the reviewer for their positive evaluation of the case presentation and for recognizing the interesting aspects of the *P. aeruginosa* population structure and observed antibiotic synergy. We appreciate the constructive suggestions to improve the manuscript, particularly regarding clarification of isolate differences, further discussion of susceptibility shifts, and consideration of additional data on phage activity and host response. Below, we provide a detailed, point-by-point response to each of these comments, including clarifications, additional discussion, supplementary experiments and additional data.

In order to enhance the manuscript, I have a number of more specific suggestions:

[Point #2]

*The differences between the *P aeruginosa* isolates are not very clear to me at least, as only % identity, an ST, an antibiogram and a gene list are provided but these are presumably four members of a diversified population of bacteria under stress. The*

“Timely bespoke bacteriophage–antibiotic combination successfully treats refractory Pseudomonas aeruginosa mediastinitis and vascular graft infection.”

CAZ-AVI and CIP synergy are interesting, as neither ab alone provides sustained growth suppression, nor do any of the phage combinations. This would be worth a little more explanation, as it bears significantly on differential phage responsiveness. Relevant aspects to discuss are bacterial defence systems and prophages in particular I think.

Good point, we checked both and added; no link with synergy BUT synergy mechanistic explanation detailed in next point below.

Modifications to the manuscript:

- Suppl. Figure S1 showing Phage Defense Systems identified in patient isolates' genomes.
- Supplementary Table S4, added prophages characterisation showing that they are devoid of VF, AMR genes or other remarkable genetic elements.
- Results section, page 12, lines 277-279. “The prophages and Phage Defense Systems (PDS) in these 4 isolates were identified and characterised (described in supplementary information). Prophages are devoid of VF, AMR genes or other remarkable genetic elements.”

[Point #3]

It would also be interesting to discuss the LEVO susceptibility shift in kinetics and contrast the mechanism of the CAZ AVI effect, since it has such a different mode of action. Do the bacterial genomic data explain this in any way?

We thank the reviewer for raising this important point. LEVO, PTZ or CZA (we note that the reviewer mentioned CIP. However, we have only used LEVO as the only fluoroquinolone) have distinct mechanisms of action, and their effects in combination with phage therapy merit deeper mechanistic discussion. We provide additional evidence of a mechanistic basis for the phage-driven antibiotic sensitisation. We used a *P. aeruginosa* MexAB-OprM receptor-deficient mutant and tested its susceptibility to our phage cocktail. The mexAB-OprM complex has previously been identified (by us and others, see references below) as a major receptor for *P. aeruginosa* phages. Its deletion leads to increased sensitivity to Fluoroquinolones, Beta-lactams and Cephalosporins. Phages targeting the MexAB-OprM complex have been used both in vitro and in vivo and achieved a phenotypic trade-off between antibiotic resistance and phage sensitivity. We tested the ability of the phages used in our therapeutic cocktail to infect this mutant. The results showed that 1 out of 3 of the phages composing our therapeutic cocktail rely on MexAB-OprM to carry out a lytic infection. These findings provide a mechanistic rationale for the antibiotic sensitisation observed in vitro and likely in vivo. We have revised the manuscript to present these critical findings (Figure 3 and supplementary Figure SXX).

References:

“Timely bespoke bacteriophage–antibiotic combination successfully treats refractory Pseudomonas aeruginosa mediastinitis and vascular graft infection.”

- Chan, B., Sstrom, M., Wertz, J. et al. Phage selection restores antibiotic sensitivity in MDR Pseudomonas aeruginosa. Sci Rep 6, 26717 (2016). <https://doi.org/10.1038/srep26717>
- Ho P, Dam LC, Koh WRR, Nai RS, Nah QH, Rajaie Fizla FBM, Chan CC, Aung TT, Goh SG, Fang Y, Lim Z, Koh MG, Demott M, Boucher YF, Malleret B, Gin KY, Dedon P, Moreira W. Screening of the PA14NR Transposon Mutant Library Identifies Genes Involved in Resistance to Bacteriophage Infection in Pseudomonas aeruginosa. Int J Mol Sci. 2024 Jun 26;25(13):7009. doi: 10.3390/ijms25137009

- Modifications to the manuscript:

- Revised Figure 4, Legend of Figure 4
- Additional Supplementary Figure S17 showing the role of MexAB-OprM in phage efficacy

[Point #4]

I think the supp data on phage-antibiotic synergy might be more strongly featured and discussed, including justification for the slightly unusual choice to combine all 4 bacteria as a mixture in testing, and to use the very high MOI of 100. Are there any data at subMIC in support of synergy?

We thank the reviewer for this insightful comment. The decision to combine all four isolates and use a high MOI of 100 was driven by the clinical context of a polymicrobial, biofilm-associated infection involving genetically related but phenotypically variable strains. This approach aimed to simulate in vivo complexity and ensure maximal killing across the population. However, we acknowledge that using a mixture of 4 bacteria is an unusual experimental design and have now revised the manuscript to explicitly justify this approach. Additionally, we have conducted additional experiments using lower MOIs, and sub-MIC antibiotic concentrations, and included these data in the revised manuscript. Our data shows a clear MIC shift for LEVO and PTZ leading to increased killing (>2-log) at sub-MIC concentration. We agree with the reviewer that our data showing phage-antibiotic synergy should be more strongly featured and discussed. We have now incorporated them in revised Figure 3 and discuss them accordingly.

Modifications to the manuscript:

- Results section, pages 15-16. Revised Figure 4.
- Included the justification of combining 4 bacteria isolates in Legend of Figure 4 in page 16 of Results section.

[Point #5]

Fig S9 shows no real suppression of growth by any phage combination beyond 6-8h, apparently using a very high MOI of 100. Are there any data at lower MOIs?

“Timely bespoke bacteriophage–antibiotic combination successfully treats refractory Pseudomonas aeruginosa mediastinitis and vascular graft infection.”

We agree with the reviewer that single-phage, and the combined-phage treatment did not lead to sustained killing over time. We provide additional evidence at a lower MOI of 1.

Modification to the manuscript:

- Additional Supplementary Figure S15

[Point #6]

Sharp initial collapses (2-3 log) in control growth kinetics are not discussed in Figs S8-14. Could the authors please comment on this and its impact on the readers interpretation of the time-kill curves.

We thank the reviewer for this important observation. The sharp initial drops in bacterial density observed in the controls (Figs S8–S14) are likely attributable to sub-optimal growth conditions for that set of experiments. Importantly, these effects were observed across both treated and untreated conditions and did not differentially impact the interpretation of phage-antibiotic synergy. We have now included this clarification in the revised figure legends in the relevant parts of the supplementary information. We have also repeated these kill kinetic experiments and showed that these initial collapses were not reproduced and did not affect the interpretation of our findings.

Modifications to the manuscript:

- New Figure 4A in Results section, page 16.
- New Figure S11-S14
- Supplementary Figure S8-S10 – The comment “The sharp initial drops in bacterial density observed in the control are likely attributable to sub-optimal growth conditions for that set of experiments” is included in every Legend of Figure S8-S10.
- Additional Suppl figure S15 (low MOI)

[Point #7]

I saw no data regarding the presence of phages in blood – do we know what clearance kinetics were like?

We thank the reviewer for this observation. We have now included additional data showing that we can detect phages DNA in the patient blood sample at day 8 and day 15 of the phage therapy course. While we were able to detect phage DNA, we are not able to quantify phage clearance in that case. We have discussed this point in our revised manuscript.

Modifications to the manuscript:

- Supplementary information. Figure S18
- Introduction section – Figure 1 (page 7) and Legend of Figure 1 (page 8 line 153 – 155)– The phrase “ (...) PT, Phage Therapy; Red blood drop symbols during PT,

“Timely bespoke bacteriophage–antibiotic combination successfully treats refractory Pseudomonas aeruginosa mediastinitis and vascular graft infection.”

indicate dates at which blood sample were taken to determine phage presence in patient blood (day 0, 8 and 15) (...).”

- Results section Page 18, lines 489-490: The statement “Phage were detected in patient blood samples at day 8 and day 15 of phage therapy by PCR using primers designed to discriminate between the three phages (Supplementary Information).”
- Discussion section, page 20, lines 567-569

[Point #8]

Do we have any evidence of phage amplification to show that phages were successfully preying on the bacterial population/s?

We thank the reviewer for raising this important point. A PCR (polymerase chain reaction) test performed on patient blood samples at day 8 and day 15 of phage therapy confirmed the presence of administered phages. While this supports phage persistence during treatment, we acknowledge that we do not have direct evidence of in vivo amplification. Demonstrating amplification was challenging given the limited number of post-therapy patient samples available. We have clarified this limitation in the revised manuscript.

Modifications to the manuscript:

- Same as previous point
- Discussion section, we have inserted “PCR confirmed phage persistence in blood at days 8 and 15, though in vivo amplification could not be demonstrated due to limited sampling.” in page 20, lines 567-569.

[Point 9]

Was there any evidence of a host inflammatory reaction to the phage administration?

The patient tolerated phage therapy well with no adverse events during treatment. Biochemically, we noted a measurable increase in the inflammatory markers; the previously normal CRP and erythrocyte sedimentation rate (ESR) increased to a peak of 16.8 mg/L, and 44 mm/h, respectively, after phage administration, as described in page 18, lines 470-473

Clinically, the patient exhibited no signs or symptoms indicative of systemic inflammation or anaphylactic responses. However, slight increases in CRP and ESR, as previously noted, alluded to a mild inflammatory response post phage administration, as described in page 18 lines 485-489

[Point 10]

Were any (targeted) bacteria or (administered) phages recovered from wounds/mediastinum/ tissue or body fluids from which we might learn about any co-

“Timely bespoke bacteriophage–antibiotic combination successfully treats refractory Pseudomonas aeruginosa mediastinitis and vascular graft infection.”

adaptation? What was done to retrieve them (eg it may be that there were no pseudomonas and no phages present soon after initial phage administration) ?

We thank the reviewer for this question. No *P. aeruginosa* or administered phages were recovered from mediastinal tissue or other body fluids after initiation of phage therapy. The absence of recovery likely reflects both the improvement effect due to ongoing targeted antimicrobial therapy and the limited sampling opportunities inherent to the clinical setting. However, PCR testing of patient blood at day 8 and day 15 of phage therapy confirmed the presence of administered phages, indicating persistence in vivo during treatment.

[Point 11]

The serum neutralization data are a little confusing as well – it seems that the serum was diluted to low levels (1%) and the bacterial density was high (in blood, a 10e9 dose will be diluted immediately to 10e4-5 etc) – it might be better to set these data aside completely or to add in more data/ revise that section significantly to make it more impactful/ informative.

We agree with the reviewer that the serum neutralisation data, as presented, are inconclusive and confusing. Based on your recommendation and aligned with Reviewer 1’s similar point, we have removed this section from the revised manuscript to avoid overinterpretation.

With regard to layout/ content:

[Point 12]

a. Fig 4 is unnecessary I think, or could move to / combine with Fig S2
We have revised Figure 4 (now Figure 5) in accordance with Reviewer 1’s comments) to illustrate the parallel tracks of Patient Journey, Clinical Workflow and Phage Microbiology workflow.

[Point 13]

b. Fig 3 could be simplified by omitting EMs and spot tests, and referring instead to Fig S3. We agree and have revised Figure 3 significantly (now Figure 4)

[Point 14]

c. Overall, the text could be greatly reduced and the focus shifted to those questions that are raised – eg is it the case that phages which are apparently poor performers in vitro can be highly effective in vivo? And to highlight the evidence for this eg phage amplification, host responses, bacterial lysis products in blood etc.

We thank the reviewer for this constructive feedback. We have significantly revised our main manuscript to highlight that phage + antibiotics are highly effective both in vitro and in vivo as either one of the modality alone is poorly effective. The critically important additional findings and modifications include:

“Timely bespoke bacteriophage–antibiotic combination successfully treats refractory Pseudomonas aeruginosa mediastinitis and vascular graft infection.”

- Results section pages 16-17, Figure 4 C-E: Antibiotic/phage synergy, MIC shift, Sub-MIC increased killing and Biofilm activity
- Supplementary Figure S17: Role of the MexAB-OprM efflux pump in phage susceptibility
- Supplementary Figure S18: Detection of Therapeutic Phages in Patient Blood

Reviewer #3

In this Case Report to the Journal, you report on a young female patient with an early refractory thoracic vascular graft infection following aortic root replacement and pulmonary homograft replacement that was successfully treated with a targeted combination of phages and antimicrobial treatment.

*This report highlights the novel use of phage therapy in a case where a re-operation with complete removal of the infected prosthesis was deemed not possible due to a history of several re-sternotomies (5 in total) and a molesting biofilm-producing bacteria (*Pseudomonas aeruginosa*). After achieving multidisciplinary consensus and following several bacteremia episodes under antimicrobial treatment, the authors opted to rapidly screen the patient for potential phage therapy and therapeutic drug monitoring. She was successfully treated with phages for two weeks, achieving inflammation control without the need for further intravenous antimicrobial treatment. At follow-up, the patient remains on suppressive oral treatment. The document is well written and the analysis as well as the conclusion are sound. From a clinical point of view, there are several questions to be answered, before accepting this interesting manuscript for publication.*

We thank the reviewer for their positive assessment of the manuscript, particularly noting the novelty of phage therapy use in this complex, non-operable vascular graft infection and the soundness of the analysis and conclusions. We appreciate the clinically focused questions raised and provide point-by-point responses below.

[Point 1]

Line 106: Why was mediastinal lavage performed 6 days after diagnosis of mediastinitis, why not earlier? And what did you perform exactly?

For clarity, we wish to emphasise that the onset of mediastinitis in this case was insidious. The patient first developed fever on post-operative day (POD) 2, initially managed as culture-negative nosocomial pneumonia. On POD 6, a significant left pleural effusion was identified and drained on POD 7. It was not until POD 10 that a complicated infection was suspected, when pleural drain output became cloudy. Due to persistent fever and *P. aeruginosa* bacteraemia commencing on POD 10, a CT thorax was performed on POD 13, confirming mediastinitis.

“Timely bespoke bacteriophage–antibiotic combination successfully treats refractory Pseudomonas aeruginosa mediastinitis and vascular graft infection.”

The diagnostic work-up was necessarily sequential, with definitive treatment instituted as new findings emerged. Management was further complicated by a concurrent left groin haematoma requiring evaluation, anaemia necessitating transfusion, the need for patient and family consent for reoperation, and emergency operating theatre waitlist constraints. The interval from diagnosis of mediastinitis to surgical intervention was three days, not six as might be inferred. Figure 1 has been revised to clarify this timeline; however, certain finer clinical details have been omitted from the manuscript to maintain focus.

Operatively, the procedure involved reopening via the prior sternotomy, removal of sternal wires, and drainage of frank pus from the sternum, which on culture yielded *P. aeruginosa*. The mediastinum was thoroughly irrigated using pulsavac, with adhesiolysis to prevent re-collection. Following irrigation, the mediastinum appeared substantially cleaner, and sternal rewiring with routine subcutaneous closure was performed. For simplicity, we have described this as a mediastinal washout. The operative details were provided by the attending cardiothoracic surgeon, who is a co-author of this manuscript.

[Point 2]

Was a tissue flap (muscle/omental) coverage of the prosthesis considered? The group of Hannover showed that this approach was feasible when treating early thoracic vascular graft infections (Umminger et al, Eur Jour Cardiothorac Surg, 2016)

There was significant pus in the mediastinum and the patient was still bacteremic at time of surgery. Tissue flap coverage was not considered by our cardiothoracic surgeons.

[Point 3]

What did you find intraoperatively, pus? Were intraoperative samples sent for culture? And if so, did they grow positive for the same microorganism?

Yes, as described above. The intra-operative pus was positive for *P. aeruginosa* and this was also described in the manuscript and presented in Figure 1 (we have revised the figure for clarity).

[Point 4]

Line 185: I agree with you that this patient was not operable due to the number of re-sternotomies and the acuity of the infection and the last surgery. However, did you consider at any point to do a re-sternotomy with mediastinal-irrigation and flap-coverage instead of re-operation on cardiopulmonary bypass and explantation of the infected prosthesis?

Various interventional and operative options were considered during multidisciplinary discussions. The cardiothoracic team, including a senior surgeon providing a second

“Timely bespoke bacteriophage–antibiotic combination successfully treats refractory Pseudomonas aeruginosa mediastinitis and vascular graft infection.”

opinion, concluded that conservative management was preferable. Following mediastinal washout and targeted antibiotic therapy, the patient demonstrated significant clinical improvement, maintained a reasonable quality of life, and expressed reluctance to undergo another major surgical procedure. As this was not the primary focus of the manuscript, these details were not included in the main text.

[Point 5]

Line 195: “Definitive surgery to remove the composite vascular graft was not possible”. Again, I agree with you that the patient is not operable, but the wording might not be precise enough, since it is always possible.

We agree with the reviewer and modified the sentence to ““Definitive surgery to remove the composite vascular graft was technically challenging with high operative risks and carried with it significant morbidity and mortality which was unacceptable to the patient.”

[Point 6]

When and how was the last follow-up performed? Did you perform an FDG-PET/ CT to control tracer uptake and steer antimicrobial treatment?

Her last follow-up was in April 2025. She remained clinically well on oral levofloxacin suppression, with no recurrence of bacteraemia. A repeat PET-CT performed at that time demonstrated further reduction in tracer uptake at the proximal ascending aorta, in the region of the aortic valve prosthesis. This image has been included in the revised Figure 2.

REVIEWERS' COMMENTS

Reviewer #1 (Remarks to the Author):

The authors have submitted a thoughtful and comprehensive revision that directly addresses the concerns raised in my initial review. The revised manuscript now presents a well-organized, clearly written, and mechanistically grounded account of a complex *Pseudomonas aeruginosa* vascular graft infection managed with a personalized phage–antibiotic combination. The addition of in vitro time-kill synergy assays, biofilm dispersion experiments, and mutant data demonstrating the role of the MexAB–OprM efflux pump as a phage receptor, and more, meaningfully establish mechanistic plausibility for antibiotic re-sensitization and demonstrate that the phage component of therapy had a rational biological basis rather than being empirically applied.

The authors have also improved clarity throughout the manuscript. The narrative flow between clinical management, microbiological work-up, and mechanistic interpretation is coherent and easy to follow. The revised figures, particularly those illustrating the phage therapy workflow and the integrated clinical–laboratory process, are well designed and enhance readability. Importantly, the authors have moderated earlier overstatements about causality and now appropriately describe the contribution of phage therapy as plausible but lean toward being ultimately circumstantial.

Some limitations inevitably remain with an $n = 1$ study. The assertion that phage therapy “restored” fluoroquinolone susceptibility, while mechanistically supported by the MexAB–OprM data, remains unconfirmed given the absence of post-treatment isolates. It may also be helpful to clarify that the detected phage DNA indicates presence rather than demonstrated replication. The discussion could be slightly condensed to maintain focus.

>>> We sincerely thank Reviewer #1 for their thoughtful and encouraging feedback. We appreciate the recognition of our efforts to address the initial concerns and to clarify the mechanistic basis for the phage–antibiotic combination therapy. Your comments regarding the manuscript’s organization, clarity, and moderate interpretation of causality are invaluable to us, and we are grateful for your detailed review and constructive suggestions, which have greatly improved the quality of our work.

In line 333- 334 of the main manuscript, the statement “Phage were detected in patient blood samples at day 8 and day 15 of phage therapy by PCR using primers designed to discriminate between the three phages” indicates presence of phage rather than demonstrated replication. We have condensed the discussion slightly further as well.

Reviewer #3 (Remarks to the Author):

Dear authors,

Thank you for addressing these points. I have no further comments.

>>> We would also like to thank Reviewer #3 for their constructive feedback and for taking the time to review our manuscript. We appreciate your positive assessment and are grateful for your consideration.